# Systematic Review on Working Mechanisms of Signaling Pathways in Fibrosis During Shockwave Therapy

**DOI:** 10.3390/ijms252111729

**Published:** 2024-10-31

**Authors:** Lot Demuynck, Sarah Moonen, Filip Thiessen, Ina Vrints, Peter Moortgat, Jill Meirte, Eric van Breda, Ulrike Van Daele

**Affiliations:** 1Research Group MOVANT (Movement Antwerp), Department of Rehabilitation Sciences and Physiotherapy, University of Antwerp, 2610 Antwerp, Belgium; lot.demuynck@uantwerpen.be (L.D.); sarah.moonen@uantwerpen.be (S.M.); jill.meirte@uantwerpen.be (J.M.); 2Department of Plastic, Reconstructive and Aesthetic Surgery, Multidisciplinary Breast Clinic, Antwerp University Hospital, 2610 Antwerp, Belgium; filip.thiessen@clinic12b.be (F.T.); ina.vrints@clinic12b.be (I.V.); 3Department of Plastic, Reconstructive and Aesthetic Surgery, Ziekenhuis Aan de Stroom, 2020 Antwerp, Belgium; 4Antwerp Surgical Training, Anatomy and Research Centre (ASTARC), Faculty of Medicine and Health Care Sciences, University of Antwerp, 2650 Edegem, Belgium; 5Department of Plastic, Reconstructive and Aesthetic Surgery, Heilig Hart Ziekenhuis, 2500 Lier, Belgium; 6Organisation for Burns, Scar Aftercare and Research: OSCARE, 2170 Antwerp, Belgium; peter.moortgat@oscare.be

**Keywords:** fibrosis, underlying mechanisms, shockwave therapy

## Abstract

Fibrosis is characterized by scarring and hardening of tissues and organs. It can affect every organ system, and so could result in organ failure due to the accumulation of extracellular matrix proteins. Previous studies suggest that mechanical forces (such as shockwave therapy, SWT) initiate a process of mechanotransduction and thus could regulate fibrosis. Nevertheless, it is largely unexamined which pathways are exactly involved in the application of SWT and can regulate fibrosis. The present article seeks to elucidate the underlying effect of SWT on fibrosis. Evidence shows that SWT activates macrophage activity, fibroblast activity, collagen amount and orientation and apoptosis, which ultimately lead to an adaptation of inflammation, proliferation, angiogenesis and apoptosis. The included articles reveal that other proteins and pathways can be activated depending on the energy levels and frequency of SWT. These findings demonstrate that SWT has beneficial effects on fibrosis by influencing the proteins and pathways. Based on these data, which highlights the underlying mechanisms, we can make preliminary conclusions about the treatment modalities of SWT in scar formation, such as the energy levels and frequencies that are necessary to prevent or treat fibrotic tissue.

## 1. Introduction

The annual incidence of major fibrosis-related diseases is a significant public health concern, with Zhao et al. reporting approximately 4968 cases per 100,000 person-years [1]. Fibrosis, characterized by tissue and organ scarring and hardening, poses a risk to all organ systems, potentially leading to organ failure through excessive extracellular matrix (ECM) protein accumulation. The pathogenesis of tissue fibrosis begins with an early inflammatory response triggered by tissue injury. This inflammation leads to the recruitment and activation of various cells, including growth factors, proteolytic enzymes, angiogenic factors and fibrotic cytokines (including transforming growth factor β (TGF β), tumor necrosis factor-alpha (TNF-α), monocyte chemoattractant protein (MCP-1), Interleukin-6 (IL-6) and IL-8). Other factors related to fibrosis resulting in ECM deposition are fibroblast growth factor 23 (FGF23), Jun N-terminal kinase (JNK), Smads, connective tissue growth factor (CTG), nuclear factor kappa B (Nf-kB), Wnt pathway [2]. Fibrotic disorders are characterized by a prolonged presence of these cells, which encourages the accumulation of tissue and the destruction of normal tissue architecture through the deposition of extracellular matrix (ECM) components [3]. As the stiffness of the affected tissue increases, oxygen diffusion is hindered, exacerbating cell damage and promoting fibrosis [3].

Recent studies have shown that controlled mechanical forces, such as those applied by shockwave therapy (SWT), can modify the fibrotic process. SWT involves brief pulses of acoustic energy that oscillate between positive and negative phases, creating a dynamic interplay of mechanical force and cavitation [4,5]. This phenomenon initiates mechanotransduction beneath the skin, a process Keuhlmann et al. define as the conversion of mechanical stimuli into intracellular chemical signals that drive gene transcription [6]. Since its introduction in 1980 for the non-invasive treatment of kidney and gallbladder stones, SWT has been increasingly applied to a variety of disorders, including musculoskeletal diseases, ischemic heart disease, neurological conditions and dermatological problems [7,8,9]. SWT stands out as an ideal therapeutic option due to its non-invasive nature, patient tolerance, ease of application, precise control over intensity and frequency, low complication rates and suitability for outpatient settings [10,11]. Its cost-effectiveness and potential application during the early inflammation phase make SWT a promising preventive therapy for fibrosis [12].

Despite the growing interest in SWT and its underlying mechanisms in fibrosis treatment, transitioning from experimental research to clinical practice poses challenges. The literature is replete with studies, primarily in vitro and animal research, exploring SWT’s effects. Nevertheless, this in vitro research has limitations relying on the lack of complexity in cell cultures, artificial conditions making it hard to predict how cells would behave in vivo, and limited timeframes that may not capture the long-term effects of therapies. Animal studies also have some limitations, such as species differences, different animal models (e.g., rodents, larger mammals) that can exhibit varying responses to treatments, and ethical concerns about the use of animals in research [13,14,15]. The rapid technological advancements in this field have only fueled further interest. This article aims to review the current understanding of how SWT influences the activation and inhibition of various cells, molecules and pathways. By examining the relationship between SWT modalities and their effects on fibrotic tissue, we seek to clarify the potential of SWT in altering fibrotic conditions. An in-depth evaluation of the literature will contribute to the development of standardized treatment protocols, influencing major cells and molecules across different fibrosis types. Highlighting the principles of fibrosis tissue repair induced by SWT underscores the importance of optimal treatment strategies. Through a synthesis of key protocol features, this review will guide the design of future SWT approaches. In this systematic review, we will focus solely on human tissue. We hypothesize that specific SWT modalities will activate distinct cells, molecules and pathways. This review begins with a brief description of the methodology, proceeds with the results and concludes with critical insights on the significant findings.

## 2. Methods

This systematic review was conducted in accordance with the 2020 updated Preferred Reporting Items for Systematic reviews and Meta-Analyses (PRISMA) recommendations [16]. The protocol for this review was registered on PROSPERO (CRD42023404145) on 1 August 2023.

### 2.1. Eligibility Criteria

The population (P) of the current review is defined as fibrosis resulting from tissue damage or/and tissue changes. The intervention (I) includes any form of SWT. The outcomes of interest must meet criteria related to any underlying mechanism, including cells and molecules, mechanisms driving fibrogenesis, mechanotransduction and pathways. No comparisons were mentioned, as the aim of this review is to examine the underlying effects of SWT in fibrosis. No additional search filters were applied. The eligibility criteria are detailed in Table 1.

### 2.2. Information Sources

A search was conducted in electronic databases by 2 reviewers in PubMed, Web of Science, Embase, and Cochrane during September and October 2022. A second search was carried out during December 2023 and February 2024. The retrieved files were uploaded in Covidence, where the 2 reviewers (LD and SM) conducted a 2-phase screening process. Although Covidence automatically removed duplicates, we also manually checked for duplicated a second time to ensure thoroughness. A detailed overview of the databases can be found in Appendix A.

### 2.3. Search Strategy

The research question was formulated using the PICO(S) strategy to develop a comprehensive and exhaustive search strategy, as outlined in Table 2. Three keywords were utilized to correspond to the components of PICO (P = fibrosis, I = shockwave and O = underlying mechanism).

### 2.4. Selection Process

Two reviewers (LD and SM) independently identified the articles that met the eligibility criteria. The screening and selection were performed in two phases using the Covidence screening tool. Initially, titles and abstracts were screened, and if the study was potentially relevant, the full text was read and assessed. During this second screening phase, the following order of exclusion was applied: language > intervention > outcome > population. Throughout this process, inconsistencies were discussed, and in cases of doubt, the opinions of a 3rd and 4th reviewers (UVD and EVB) were considered (see Figure 1).

### 2.5. Data Items and Collection

The two reviewers (LD and SM) extracted the necessary data from the remaining articles. They collected Information on the underlying mechanism of interest, fibrosis type, demographics, publication information (author, publication date, study design), intervention (treatment modalities and timing), evaluation (type and timing) and also outcome measures. The first reviewer (LD) completed the evidence table (see Table 3), and then the second reviewer (SM) independently reviewed it for accuracy and completeness.

### 2.6. Risk of Bias in Individual Studies

The internal validity of each study was assessed using appropriate risk of bias (RoB) tools tailored to the study’s design. For randomized control trials (RCTs), the ROB-II checklist was employed, which evaluates five domains crucial for determining a study’s overall methodological quality and its ‘intention to treat’ effect. These domains include the randomization process, selection of participants, deviations from classification and intended interventions, missing outcome data, measurement of the outcome, and selection of the reported result. Outcomes on the ROB-II can be classified as ‘low’ or ‘high’ risk of bias or may indicate ‘some concerns’ [34].

For non-RCTs, the ROBINS-I checklist was used, assessing seven domains that can be rated as critical, serious, moderate, or low risk of bias. The first two domains focus on confounding factors and the selection of participants at baseline. The third domain evaluates the classification of interventions, while the final four domains examine issues arising after the intervention [35].

The overall level of evidence for each study was assessed using the GRADE [Grading of Recommendations, Assessment, Development and Evaluation] scale. Conclusions were provided for each outcome measure, with an overall rating of the quality of evidence being determined by taking the lowest quality of evidence. The GRADE scale categorizes evidence into 4 levels: very low (indicating that the true effect is probably very different from the estimated effect), low (suggesting that the true effect might be very different from the estimated effect), moderate (meaning the authors believe that the true effect is probably close to the estimated effect) and high (where the authors are highly confident that the true effect is similar to the estimated effect) [36].

Different RoB tools were used due to the inclusion of both RCTs and non-RCTs, following specific guidelines for interpreting the GRADE tool.

The two reviewers (LD and SM) independently assessed the RoB without knowledge of each article’s details to identify disagreements or the need for further discussion. The results were then compared and in instances of disagreement, prompted re-analysis of the article to reach consensus.

## 3. Results and Discussion

### 3.1. Study Selection

Because of the large number of hits, we will focus solely on the studies of human outcomes in this review. A total of 216 articles were included for a second screening, from which 143 articles were considered eligible based on the inclusion and exclusion criteria, comprising 29 human studies and 114 animal studies. An overview of the study selection across different databases is displayed in Figure 1.

### 3.2. Study Characteristics

During the first screening phase, we encountered disagreements in 2.1% of the cases, and during the second screening phase, disagreements occurred in 25.77% of the cases. In instances of disagreement during the second screening phase, we consulted two additional reviewers (UVD and EVB). One was RCT, and 28 were non-RCT. Table 3 gives a detailed overview of the articles that were included. The primary reasons for exclusion included inappropriate patient populations and irrelevant outcomes. Specifically, the study population needed to exhibit some form of fibrosis; however, several trials incorporated healthy tissue to analyze the effect of shockwave therapy. At the outcome level, some studies assessed fibrosis through subjective evaluations, which did not meet our criteria.

### 3.3. Risk of Bias in Studies

Quality assessment was conducted using the ROB-II tool for RCTs and the Robins-I tool for non-RCTs. The ROB-II assessment focused on evaluating the quality of (specific aspect or aspects of the studies), with the results presented in Appendix A (see Appendix A). For non-RCTs employing a quasi-experimental design, the ROBINS-I tool was used. The risk of bias was evaluated on the outcome level, more specifically on the fibrotic phase and relating underlying mechanisms. The majority of the outcomes (N = 44) demonstrated a low risk of bias (RoB), while three outcomes were found to have some concerns, and 11 outcomes had serious RoB. The major reason contributing to an increased RoB was the measurement of outcomes. Additional, though less frequent, factors included the selection of participants, classification of interventions and missing data.

The timing of assessments was often not specified. Sometimes, inadequate descriptions of methodology and population characteristics in many articles hinder the ability to assess the risk of bias and study quality effectively. The variation in methodologies for molecular investigation could confound any statistical associations, either in favor of or against the trial hypothesis. Conceivably, the use of ‘better’ methods of determining molecular alterations and optimized tissues (biopsy vs circulating) in carefully conducted trials with rigorous sampling and storage conditions and sufficient follow-up with many longitudinal samples, even if not of large size, can provide good evidence of predictive and prognostic significance. Consequently, certain outcomes were challenging to compare, and in some instances, only a single article addressed a specific outcome. This often resulted in the assignment of predominantly low GRADE scores (see Table 4).

### 3.4. Results of Individual Studies

#### 3.4.1. Study Population

Figure 2 provides an overview of the included tissues in human fibrosis. Most studies specified their tissue and fibrosis-related aspects. Some articles performed in vitro [3,4,5,6,8] studies from the affected fibrotic samples.

Skin fibrosis has been most commonly associated with wounds, e.g., scarring caused by chronic ulcers [3,9,10] and elective surgery [11,12]. Other skin scarring studies have focused more on the scar type, more specifically, keloid scars [18], excision wounds [8], retracting hand scars [19] and postburn hypertrophic scars [4,5]. The other two articles dealt with more chronic diseases that are associated with fibrosis: systemic sclerosis [20] and adipose tissue in obese patients [17].

In bone tissue, three articles described the effect of osteoarthritis on osteoblasts [23], chondrocytes [24] and bone stromal cells [37]: one on the knee and two on the femoral head, respectively. In addition, two articles included patients with osteonecrosis, both from the head of the femur [25,26]. Cells without tissue specification were fetal osteoblastic cells [40] and mesenchymal stem cells [21]. One article was interested in tissue after spinal elective surgery [27].

The Achilles tendon was the focus of two tendon/muscle articles [28,29], while another study investigated shockwave application on the anterior cruciate ligament [38]; all these tissues are prone to fibrosis due to previous trauma.

The researchers who were interested in internal fibrosis assessed the lungs, heart, organs of the urinary system and the immune system. For the lung, Di Stefano et al. focused on the effect of bronchial fibroblast in chronic obstructive pulmonary disease (COPD) patients [30]. Ischemic heart disease was the pathology of interest for the underlying mechanisms of shockwave (SW) [6,31,32]. In the urinary system, prostatic carcinoma cells were involved (after a prostatectomy) [33], and finally, in the immune system, the Jurkat T-cells were investigated [28]. The sample size ranged from 5 to 75 human individuals; most studies included 5–39 individuals, except for one study that included 75 [3] individuals.

#### 3.4.2. Treatment Modalities of Shockwave Therapy

The intervention identified in the included studies consists of shockwave therapy. The protocols in these studies vary in treatment modalities. More specifically, energy flux density (EFD), intensity and frequency were reported. Additionally, the duration of each treatment, treatment amount and frequency/week were also important modalities that we considered. Some studies compared one SWT treatment with various other treatments (e.g., steroid injections and hyaluronic acid). Studies have compared different treatment modalities and the timing of treatment. In the included studies, treatments were carried out with an EFD between SWT: 0.0024–0.32 mJ/mm^2^, frequency: 3–5 Hz and pulses ranging from 150–3000.

The effect of the SWT on the underlying pathways and cells depends on the EFD (see Figure 3). Firstly, growth factors (GF) can have a positive impact on alpha-smooth muscle actin (αSMA) with an EFD of 0.32 mJ/mm^2^ and reaching a total of 1000 pulses with a peak positive pressure of 90 MPA each treatment [11]. Tinazzi et al. used an EFD of 0.2–0.25 mJ/mm^2^, and after 1000 shots with 4 Hz, they showed a significant increase in vascular endothelial growth factor (VEGF) [20]. Secondly, also Rinella et al. found a positive effect on integrin α11 with an EFD of 0.32 mJ/mm^2^ and 1000 pulses [11]. In the study of Cui et al., two different EFDs were used, and they were interested in the change of fibronectin (mRNA and protein). The assessment after 24 h found a significant decrease in all regimens (0.03, 0.30 and 0.10 mJ/mm^2^), whilst after 72 h, only an EFD of 0.03 and 0.1 mJ/mm^2^ showed a significant decrease of fibronectin messenger ribonucleic acid (mRNA) [5]. Contrary to fibronectin mRNA, the fibronectin protein was significantly increased after 24 and 72 h. Suhr et al. used focused SWT, and after delivering 0.2 mJ/mm^2^ for 30 min, F-actin showed remarkably reduced disorganized F-actin [37]. In this case, SWT is used to address collagen I production, and two studies by Rinella et al. found a decrease in collagen I when patients were treated with 1000 pulses of 0.32 mJ/mm^2^ [11,33]. Similar to Rinnella et al., a significant decrease was uncovered by Wang et al., who treated their patients with an EFD of 0.11 mJ/mm^2^ and reached 500 impulses with four shocks/s. Their patients received three SWT treatments in 6 weeks [18]. Conflicting evidence exists regarding the effectiveness of SWT on collagen II. In the study of Suhr et al., no differences between the treatment and control groups were found, whereas Leone et al. discovered an upregulation of collagen II [29]. In the study of Vetrano et al., a shift was shown from collagen I to collagen II (regulated by enhanced SRY-related HMG box (SOX9)), and the dosage of SWT was 0.14 mJ/mm^2^ and 1000 impulses [24].

A significant decrease in collagen III was found when the patients were treated with an EFD of 0.11 mJ/mm^2^, 500 impulses and four shocks [15]. As mentioned, Wang et al. treated their patients three times a week for six weeks. Furthermore, a decrease in collagen V was found by Rinella et al. [10]. Lastly, Tinazzi et al. found a significant increase in intracellular adhesion molecule-1 (ICAM-1), membrane cofactor protein-1 (MCP-1) after a treatment with a dosage of 0.20–0.25 mJ/mm^2^, reaching 1000 pulses with 4 Hz [34].

Inflammation phaseDuring inflammation, treatment modalities in terms of EFD ranged between 0.0024 and 0.30 mJ/mm^2^ [9,30,31,32]. Only one study described higher energy levels of 180 mJ, but this was in preoperative tissue [17]. This study described higher energy levels and their beneficial effects on natural killer (NK), T, B lymphocytes and proinflammatory macrophages. The lowest EFD (0.0024–0.09 mJ/mm^2^) and 200 pulses showed positive effects on chemokine ligands 1, 2 and 3 (CXCL1, 2 and 3) [21];On macrophages (M1 and M2) and their associated cytokines, interleukin 1 and 6 (IL-6, IL-1) and tumor necrosis factor α (TNF-α), chemokines and VEGF, studies used EFD between 0.11–0.14 mJ/mm^2^ during the inflammatory phase. In bronchial fibroblasts, a higher EFD of 0.30 mJ/mm^2^ was used with the same pulse frequency of 500 was used [30];Proliferation phaseDuring the proliferative phase, the EFD reported in the included studies also varied generally from 0.0024 to 0.30, but other cellular changes have been reported. As mentioned above, Wang et al. used low EFDs between 0.0024 and 0.09 with 200 pulses and found effects on mitogen-activated protein kinase 9 (MAPK9). More or less similar results were found by Weihs et al., using EFDs of 0.030–0.19, 3 Hz and 10–300 pulses, also finding an effect on Akt and extracellular-regulated kinase (Akt/Erk) pathway besides p38 MAPK [8]. In another study, an EFD of 0.25 mJ/mm^2^, 3000 pulses and 3 Hz was used on the Erk pathway. Proliferating cell nuclear antigen (PCNA) and fibronectin were positively affected by an EFD of 0.11 mJ/mm^2^ [31].In 4 other studies, EFDs between 0.15 to 0.18 and pulses between 250 and 5000 were used. Positive effects were found on cell viability, cell proliferation and cell migration through changes in collagen-I α, TGFβ and VEGF [38]. Similar results were found for differentiation markers (alkaline phosphatase, core binding factorα (Cbfα): glyceraldehyde-3-phosphate dehydrogenase (GADPH), osteocalcin (OCN) β-actin, peroxisome proliferator-activated receptor gamma (PPAR)). In the expression of T cells, Yu et al. found a higher expression of IL-2 with an EFD of 0.18 mJ/mm^2^ [7]. Three studies demonstrated EFD dependence in cellular changes in p21, p27 and Notch 21, finding significantly higher only by 0.2–0.3 mJ/mm^2^ EFD [5]. Another study showed that Ki67 varies with EFD. It was only at 0.2 mJ/mm^2^ that they found a significant increase [37]. In the meantime, Leone et al. found a significant increase in Ki67, but they did not specify the EFD. However, no significant change in Ki67 was found in another study [29];Activation and differentiation phaseIn keratinocytes, two studies by Cui et al. used different EFDs and found different outcomes depending on EFD. They were interested in keratin 6.17 and keratin 1.10, and the results showed a significantly higher amount of keratin 6.17 by an EFD 0.2–0.3 and 4 Hz, respectively, while keratin 1 and 10 were not significantly changed at 0.10 mJ/mm^2^, but at 0.20–0.30 they were significantly higher [4]. Zhai et al. found higher Cbfα1 gene expression at 0.16 mJ/mm^2^ EFD [26]. An EFD of 0.16 mJ/mm^2^ also significantly decreased miR138 [21];ApoptosisDuring apoptosis, different studies indicate EFD between 0.11 and 0.30, 4 to 5 Hz and 150 to 500 to 1000 pulses [5,9,37]. Most of the apoptosis was described in the caspase pathway, and a significant decrease of 0.2 mJ/mm^2^ and an increase of 0.3 mJ/mm^2^ was found [37]. Similar results were found for the proapoptotic factor Bcl-2 associated X-protein (Bax) and the anti-apoptotic factor B-cell lymphoma (Bcl) [5].

#### 3.4.3. Assessment Protocol

All the included studies performed a histopathological assessment to analyze the underlying mechanisms and regeneration of fibrosis. All the included studies assessed with objective analysis of the underlying cellular and molecular mechanisms.

Hematoxylin and Eosin (H&E) staining was used by 1 study [8] to visualize the tissue morphology. Other studies frequently performed an immunohistochemistry staining (IHC) [8,17,30,37] or an immunofluorescence staining [11,24,29,32,33,38] for the detection of proteins. One study did not exactly specify the staining [26]. Two studies specified their IHC and used Masson’s trichome staining [9,31]. Wang et al. used the immunoblotting for the separation of proteins [41]. Western blots [WB] were applied for the mRNA and protein expression in eight studies [4,5,7,8,21,26,32,33]. Enzyme-linked immunosorbent assay (ELISA) was used in four studies [20,24,30,42], and Tinazzi et al. used a blood sample as well. A flow cytometry was performed in two studies [6,23].

Molecular biology was frequently measured by polymerase chain reaction (PCR). More specifically, reverse transcription PCR (RT-PCR) [3,24,25,26,29,30,32,37,42], real-time PCR [4,5,21,39] and quantitative PCR (qPCR) [28,31] were measured. The detection of apoptotic cells was assessed through a terminal deoxynucleotidyl transferase (TdT) mediated dUTP nick-end labeling (Tunel) assay; only one study used this Tunel assay [27].

#### 3.4.4. Underlying Mechanisms Caused by SWT

The effect of SWT on fibrosis is not straightforward. For this reason, the underlying mechanisms in human fibrosis were divided into different wound/scar healing phases, with regularly overlapping findings. We made a subdivision into different phases with the corresponding processes, see Figure 4. Consequently, distinct tissues were ignored because we discovered similar underlying mechanisms in the different organ tissues, see Table 3 and Table 5.

##### Underlying Mechanisms Caused by SWT in Different Phases of Fibrosis

Various studies have explored the effect of SWT on different cell types implicated in fibrosis, including fibroblasts, myofibroblasts and collagen, without specifying the exact precise fibrotic phase. Rinella et al. focused on myofibroblasts, which typically exhibit high contractility and low migration potential. They observed increased migration and reduced contractility in response to SWT, suggesting a potential role in facilitating fibroblast migration to damaged sites and promoting granulation tissue formation. Additionally, Rinella et al. noted elevated expression of αSMA, a marker associated with myofibroblast activation, and a significant reduction in integrin α11, a key mediator of mechanotransduction [11].

Another study explored fibroblast behavior, particularly in fibronectin-rich tissue, at both mRNA and protein levels, yielding conflicting results. While mRNA levels of fibronectin decreased, protein levels increased, suggesting complex regulatory mechanisms [5]. Two studies investigating bone tissue examined the effects of SWT on β integrin and cluster of differentiation 29 (CD29) without significant findings. However, another study demonstrated a significant reduction in disorganized F-actin, indicative of cytoskeletal remodeling, following SWT [22].

Excessive and disorganized collagen, a hallmark of fibrosis, was evaluated in multiple studies, with most reporting decreased levels of collagen types I, III, and V. Type I collagen exhibited significant reduction in two studies [1,5,18] while the others reported decreases without statistical significance [11,33]. Similarly, significant decreases in type III collagen were observed in one study [18]. Additionally, upregulation of collagen-Iα1, a subunit of type I collagen, was noted after SWT [38]. Type II collagen levels were inconclusive, with one study reporting no difference and another reporting upregulation [22,29]. SOX9, a transcription factor regulating type II collagen, was examined, showing variable effects. Rinella et al. found a decrease in collagen type V [11].

Bone morphogenetic protein 2 (BMP-2) and RUNX2, crucial transcription factors in bone development, were found to be significantly increased [25]. Tinazzi et al. investigated various fibrosis markers, such as endothelial adhesion molecules (ICAM-1, MCP-1), circulating endothelial cells, endothelial progenitor cells, and VEGF, all significantly elevated except for nitric oxide (NO), which increased but not significantly in all patients [20].

##### Effect of Shockwave Therapy on Inflammation

Inflammation can be suppressed by the application of SWT, as demonstrated in various studies, which are explained by using different markers and describing different cellular responses.

Firstly, in fibroblasts, CD117, PCNA, CD90, TGF β1, procollagen I, and NF-kpB-65 were investigated [30,44]. Di Stefano et al. made a distinction between mRNA and protein for the TGFβ1, PCNA and CD117 markers. PCNA mRNA was not increased in the ESW group, but PCNA protein had a tendency to increase. No significant change was found in the TGFβ1 protein and in mRNA, while in the protein, a significant increase was found. In procollagen I, no significant difference between the treatment and control groups was found. For NF kB-65, a significant decrease was observed. Significant higher expression was found for Yes-associated protein 1 (YAP1) and higher levels of αSMA [10,27].

Secondly, two articles investigated the effect of SWT in macrophages by using different markers: cytokines and growth factors [9,17]. SWT has a positive effect on the pro-inflammatory CD68 and CD163. In 60% of the patients, Holsapple et al. explained a significant decrease in CD68, which is related to a higher expression of the marker from M2 macrophages (CD163) compared to the marker from M1 macrophages (human leukocyte antigen (HLA-DR), SOC3) [9]. This leads to an overexpression of anti-inflammatory macrophages. The number of macrophages in the treated subjects was significantly higher than in the control group. However, the type of marker and macrophage determined the positive effects in the aforementioned studies.

The expression of cytokine and growths in macrophages was studied by Holsapple et al.: respectively, IL-6 and TNF, TGF and VEGF. They found a significant increase in TNF, IL-1, platelet-derived growth factor (PDGF) and TGFβ by an intensity of 150 pulses. At 300 pulses, only a significant increase in TNF and TGFβ was observed [9]. However, there was no significant change in IL-6 and VEGF. Nevertheless, Wang et al. found in keratinocytes a small but not significant change in these last cytokine and growth factors [18].

On the other hand, two studies explained the effect of anti-inflammatory cytokines in cells other than macrophages. In proteins, TGFβ1 and IL-6 [3], IL-10 [24] and in mRNA, a CD44 marker [24] were assessed. These studies found a significant increase in IL-10 and CD44 markers [24]. Moreover, one of these two studies also explained the effect on the pro-inflammatory cytokines, and they found a significant reduction in TNFα, IL-6 and IL-17 [24].

Cyclooxygenase 2 (Cox-2) is an enzyme that can induce growth factors and thus stimulate the tissue response [17]. In the Cox-2 epidermis, it was expressed in keratinocytes, and a diffuse pattern was found in the treatment group and a focal pattern in the control group. In the dermis, the presence of fibroblasts and inflammatory cells positive for Cox-2 was observed in the treatment group compared to the control [17].

In lymphocytes, the marker CD20 in B lymphocyte [35] and the marker CD3 in T lymphocyte [35] showed a higher expression, which is related to an early inflammatory response. Natural killer (NK) cells, a group of effector lymphocytes, were higher in comparison with the non-treated side [17].

Wang et al. elucidated chemokine pathways, highlighting the impact of Shockwave Therapy (SWT) on ligands such as CXCL1, CXCL2, and CXCL3. Their findings revealed elevated levels of these ligands following SWT. Additionally, they observed a decrease in MAPK-9, indicating modulation of the MAPK signaling pathway [31].

This study showed that the overexpression of integrinα11 was reduced. Due to the overexpression of integrinα11, cells become less sensitive to ESW in terms of αSMA expression, cell contraction and migration. A role for ITGα11 in the translation of SWT signaling in human adipose-derived stem cell (hASC) responses was also suggested [11].

##### Shockwave Therapy Increases Angiogenesis

Eight different studies investigated the effect of SWT on angiogenesis in various cell types and markers associated with angiogenesis. Across these studies, diverse clusters of differentiation were examined, including CD14, CD34, CD105, CD31, CD9, CD63, and CD163 [4,6,10,11]. Interestingly, while CD34 showed no statistically significant changes in myofibroblasts [11,17], both CD105 and CD31 demonstrated markedly increased expression levels. Notably, CD105 elevation was observed only at higher EFD in the study by Iannone et al., contrasting with findings from Modena et al. Conversely, Gollman et al. unveiled an inhibitory effect on mRNA translation attributed to CD9, CD63, and CD81 mediated by miR-19a-3p [6]. Moreover, a consistent upregulation of angiogenic markers such as VEGF, VEGFA and VEGFR-2 was noted, suggesting enhanced angiogenic potential during early scar processes [3,17,25,32,45]. Lastly, SWT resulted in an amelioration of the laminin/integrin ratio in fibroblasts.

##### Effect of Shockwave Therapy on Proliferation

Proliferation is a crucial phase in fibrosis, which can lead to cell regeneration and cascades of cellular changes and pathways. First of all, PCNA is a protein that plays a critical role in deoxyribonucleic acid (DNA) replication. The DNA repair process was slightly changed, yet not significantly. SWT should be able to proliferate cells due to this PCNA upregulation. Two studies assessed the effect of SWT on growth factors, in particular on TGFβ, VEGF and VEGF A and B and unraveled an increase of these proteins during proliferation. Ki67 is known as an important proliferative marker and tissue development, but several studies found conflicting results depending on the EFD [9,22,24]. Suhr et al. discovered a significant increase after 6 h and 12 h during treatment of an EFD 0.2 mJ/mm^2^. On the other hand, they did not find a significant effect after 6 h with an EFD of 0.3 mJ/mm^2^, while after 12 h they did find a significant decrease. In addition, Vetrano et al. did not find any changes after SWT, but two other studies explored an increase in Ki67 [8,24,29]. Only one of these studies found a significant increase [29].

Moreover, studies frequently investigated enzymes involved in various cellular processes and signaling pathways, such as MAPK, which facilitates signal transmission from extracellular stimuli to intracellular targets. In this review, the included studies focussed on P38, P42, P44 and MAPK-9. P38 MAPK showed enhancement [8,43], while MAPK-9 was found to decrease [31]. For P42 and P44, we found inconclusive results. One study examined mitogen-activated protein kinase 1 (MEK1) and MEK2 proteins together within cells; these dual-specificity protein kinases function in MAPK/Erk pathways [8]. Weihs et al. found a downstream regulation, so the pathways transduce extracellular signals to downstream targets, ultimately leading to the regulation proliferation [8].

Another pathway that was examined in different studies was the Akt/Erk and Erk ½ pathway. Akt and Erk pathways often crosstalk and regulate each other’s activity through various mechanisms; Weihs et al. explored a downstream regulation of this pathway [8]. SWT can adapt the Erk halfway through VEGF and VEGF-R, and thereby, an enhancement was perceived by inducing the release of adenosine triphosphate (ATP) [8].

Keratins are crucial structural proteins primarily found in epithelial cells, where they form intermediate filaments providing mechanical support. Depending on the EFD, significant reductions in keratins 5 and 14 were observed [4].

Other important regulators in fibrosis proliferation and homeostasis include p21, p27, and Notch1. An EFD of 0.1 mJ/mm^2^ did not affect these regulators, but at 0.2 and 0.3 mJ/mm^2^, results varied depending on the assessment timing: levels were significantly higher at 24 h and lower at 72 h [4]. Vetrano et al. reported a significant decrease in p16 following SWT [29].

Studies have seldom examined isoenzymes involved in the proliferation process. Zhai et al. focussed on alkaline phosphatase, CBFα, B-actin, OCN and PPARϒ, all of which are elevated following SWT compared to the control group, except for PPARϒ, which remained steady in both groups across all time points [26].

Subsequently, SWT effected fibroblasts during the proliferation by increasing water-soluble tetrazolium salts (WST-1) and membrane-proximal external region (M-PER) [30]. In immune T-cells, higher levels of PX27 receptor FAK activation and MAPK were observed with impulses up to 250, plateauing at baseline with 350 impulses. Yu et al. reported an enhancement of cytokine interleukin 2 (IL-2) [7].

Lastly, ECM remodeling and cell adhesion markers matrix metalloproteinases 1 and 2 (MMP 1 and 2) were significantly reduced [28]. α/βMHC, FVIII and GADPH showed increased expression due to SWT contributing to overall proliferation modification.

##### Effect of Shockwave Therapy on Activation/Differentiation Markers

These markers typically denote molecular or cellular characteristics indicating the activation or transformation of cells involved in the fibrotic process. In a study by Cui et al., keratin, a fibrous structural protein crucial in epithelial tissues, was examined [4]. Under normal conditions, keratins exhibit high resistance to mechanical stress, but in fibrosis, this resistance diminishes. Epithelial tissues such as the skin are predominantly composed of keratinocytes, which produce keratin proteins, forming a cytoskeletal network that provides mechanical stability and stress resistance.

The impact of shockwave therapy (SWT) on keratin expression was investigated in two studies by Cui et al., which focused on the dynamic response of keratin subtypes to this therapeutic intervention. Specifically, the studies examined keratin 6, 17, 1, and 10, which are components of epithelial tissues with diverse roles in structural integrity and barrier function [4].

The p21, p27 and Notch are proteins involved in cellular signaling and regulation that play a critical role in activation and differentiation in physiological processes such as fibrosis. Depending on the energy flux density, different results were found: at 0.1 mJ/mm^2^, these proteins were not significantly changed, whereas at higher EFDs of 0.2–0.3 mJ/mm^2^, they were significantly higher both after 24 h and significantly lower after 72 h [4].

Signaling pathways involved in the regulation were investigated by Hu et al.: they found that the FAK/Erk1/2 signaling pathway, which regulates cell adhesion and migration, was down-regulated after SWT [21]. RUNX2, a transcription factor crucial for skeletal development and osteogenesis, was observed to increase following SWT, facilitating cellular differentiation. Another transcription factor, Cbfα1, is involved in osteogenesis regulation, often forming complexes with RUNX1, showing conflicting results with one study reporting a significant increase in gene expression but a decrease in protein expression [26]. Leone et al. studied the effect of SWT on αSMA, observing increased levels post-treatment consistent with findings by Rinella et al., albeit in different fibrosis stages [29]. Finally, Hu et al. examined miR-138, a molecule pivotal in gene expression regulation, demonstrating a significant decrease following SWT application [21].

##### Effect of Shockwave Therapy on Apoptosis

Several studies have described a pro-apoptotic effect of SWT, with most studies describing the caspase pathways and their associated ligands and receptors [4,22,46]. 2 studies investigated the caspase-3 pathway, Wang et al. found small but no significant alterations, whereas Suhr et al. found different findings depending on the EFD [18,22]. At 0.2 mJ/mm^2^, they found a significant decrease, and at 0.3 mJ/mm^2^, they found a significant increase. Cui et al. were interested in other caspase pathways, more specifically, the caspase 14, Bax and Bcl2 receptors. Caspase 14 was not significantly different at 24 h, but after 72 h, it was significantly different. Bax (a pro-apoptotic factor) and Bcl2 (an anti-apoptotic factor) were significantly higher after SWT [4]. However, they were still lower than in untreated cells. Other studies investigated the effect of SWT on signaling pathways. Holsapple et al. found an increase in Erk/Akt signaling but no changes in pAKT/totalAKT and pSTAT3/totalSTAT3 [9]. Another apoptotic factor was examined by Cui et al.: Ask1 and this was found to be significantly higher after SWT [4].

### 3.5. Discussion

#### 3.5.1. General Aim

In general, fibrosis results from chronic inflammation and tissue damage, leading to collagen fiber deposition and scarring, affecting organs such as the lungs, liver, heart and skin. Fibrosis treatment can include several treatments, whereby SWT can be seen as an additional therapy. SWT is not the standard of care in fibrosis, but relying on these results, it could be necessary to include SWT in the fibrosis treatment to control the cellular and molecular mechanisms. Treatment of fibrosis can, next to SWT, consist of invasive (e.g., surgery), non-invasive therapy (e.g., in burn scars) or pharmacological treatment (e.g., lung fibrosis). SWT can be a helpful treatment before or post-surgery to reduce pain [47,48,49]. Nevertheless, conventional treatments have limitations, prompting researchers to explore alternative regenerative therapeutic approaches like SWT. Comprehensive understanding is crucial for preventing or treating fibrosis and understanding the effects of shockwaves. SWT, defined as mechanotherapy, dynamically regulates inflammation, proliferation, activation and differentiation in various tissues impacting cells, such as fibroblasts, growth factors, collagen, macrophages and keratinocytes [4,5,50].

The aim of this systematic review is to elucidate SWT’s effect on the underlying mechanism in human fibrosis, exploring different phases and cell types involved in fibrotic processes. This review offers new insights into shockwave therapy’s role in coordinating and regulating fibrotic tissue. However, significant challenges and unanswered questions remain. Optimizing treatment parameters, such as shockwave energy and dosage, is crucial to maximizing therapeutic benefits while minimizing adverse effects.

#### 3.5.2. Most Important Findings

This review highlights significant points regarding the use of shockwave therapy (SWT). Based on the study results, an Effective Energy Flux Density (EFD) between 0.0024 and 0.32 mJ/mm^2^ is recommended. SWT mechanically disrupts fibrotic tissue, alleviating tissue stiffness and promoting tissue remodeling.

During the initial inflammation and subsequent proliferation phases, even a minimal EFD as low as 0.0024 mJ/mm^2^ can induce cellular and molecular alterations. Higher EFDs ranging from 0.11 to 0.30 mJ/mm^2^ are required during differentiation and apoptosis. Multiple studies emphasize the importance of time-dependent assessments and energy flux-dependent treatments in managing fibrosis.

Mechanotransduction, the process through which cells convert mechanical stimuli into biochemical signals, is a critical mechanism underlying the effects of shockwave therapy (SWT) on fibrotic tissues. The reviewed studies demonstrate that SWT exerts its therapeutic effects through several mechanotransduction pathways that influence cellular behavior, extracellular matrix (ECM) remodeling, and inflammatory responses, all key factors in fibrosis management. The observed effects of ESWT on diverse cells and pathways are noteworthy. Firstly, SWT stimulates local inflammatory processes by mediating and modulating cellular and molecular proteins and signaling pathways. Fibroblasts differentiate into myofibroblasts characterized by high αSMA expression [6]. SWT can modulate fibrosis by reducing fibroblast quantity. Integrins, particularly Integrin α11, play a crucial role in mechanotransduction, influencing tissue organization and inflammation. Overexpression of Integrin α11 in cells reduces their sensitivity to ESWT regarding αSMA expression, cell contraction, and migration [12]. The Yes-associated protein (YAP) and transcriptional coactivator with PDZ-binding motif (TAZ) are mechanosensitive transcription factors activated in response to mechanical stress. Elevated YAP activity is commonly associated with tissue stiffness and fibrosis [51]. YAP responds to the skin fibroblast’s physical environment and regulates cell proliferation, ECM deposition, and remodeling. Mechanotransduction during SWT not only influences fibroblast activity but also impacts macrophage polarization. Studies have shown that low-intensity SWT can shift macrophage populations from a pro-inflammatory (M1) phenotype to an anti-inflammatory (M2) phenotype, largely through the activation of the ERK 1/2 pathway [9,20]. This transition is mediated by changes in the expression of cytokines (e.g., TNF-α and IL-1) and growth factors such as PDGF and TGF-β.

Furthermore, studies suggest implications for inflammation and tissue repair processes mediated by chemokine signaling. SWT stimulates inflammation modulation and growth factors, promoting tissue regeneration through cytokine expression [18]. Higher expression levels were also observed in NK cells and lymphocytes, indicating a role in reducing inflammation and aiding in healing and recovery.

Secondly, fibrosis involves excessive accumulation of aberrant extracellular matrix (ECM). Collagen, crucial for tissue strength and resilience, plays a central role in fibrosis by excessively depositing ECM components, particularly type I and type III collagen fibers, leading to scarring and organ dysfunction. SWT has been shown to replace collagen type I with collagen type III.

Thirdly, SWT affects wound edges and cell contraction, involving αSMA and TGF-β [11]. Endoglin, a transmembrane glycoprotein, acts as a coreceptor for TGF-β ligands, crucial for angiogenesis and inflammation regulation. Macrophages, marking lesion chronicity, facilitate inflammation to proliferation transition. ESWT stimulates angiogenesis, which is crucial for tissue repair through enhanced perfusion, oxygen and nutrient delivery. Angiogenesis plays a pivotal role in the fibrotic process, particularly in conjunction with inflammation, which is a key driver of fibrosis. During tissue injury, inflammation triggers the release of pro-inflammatory cytokines and growth factors, such as VEGF (vascular endothelial growth factor), which are critical to both fibrosis and angiogenesis. These factors stimulate the formation of new blood vessels, which in turn supply oxygen and nutrients to the fibrotic tissue, sustaining its growth. However, the aberrant regulation can exacerbate fibrosis, as excessive vascularization can promote further inflammation and fibroblast activation, leading to increased extracellular matrix (ECM) deposition. In the context of shockwave therapy (SWT), recent studies suggest that SWT modulates angiogenesis by influencing the expression of key angiogenic markers such as VEGF and CD31. This modulation could potentially play a therapeutic role in fibrosis by balancing angiogenesis to facilitate tissue repair without triggering excessive ECM deposition. In fibrotic tissues, controlling angiogenesis may also help limit the progression of fibrosis by reducing the oxygen supply to fibroblasts and attenuating inflammatory signals [52].

The interplay between inflammation, angiogenesis, and fibrosis suggests that targeting angiogenesis during the inflammatory phase of fibrosis could be a viable therapeutic strategy. By applying SWT at appropriate energy flux densities, it may be possible to modulate angiogenesis and thereby influence the fibrotic process in a beneficial manner. Understanding this relationship will be crucial in optimizing SWT protocols for fibrotic conditions.

Additionally, during proliferation, studies on Ki-67 show varying results depending on EFD. Keratin, an intermediate filament protecting epithelial cells and strengthening skin, fluctuates post-SWT, with heightened levels observed transiently within 24 h and decreased levels after 72 h [4,5].

Finally, consistent with previous studies, ESWT significantly controls apoptosis via caspase pathways. These findings underscore the time-dependent and energy flux-dependent assessments and treatments required for fibrosis.

Surprising results between the fibronectin protein and mRNA results were found; the results of Cui et al. were based on the same type of scar tissue, specifically post-burn hypertrophic scars, where we expect the same scar healing in both type and duration. 

Fibronectin, an extracellular matrix protein, is often upregulated during epithelial-to-mesenchymal transition (EMT), a key pathological process in fibrosis where epithelial cells acquire mesenchymal characteristics. One possible explanation for these unexpected results is that fibronectin mRNA may respond earlier to stimuli (within 24 h) compared to protein synthesis, which could be delayed or sustained over a longer period. Another factor could be the differential localization of the mRNA and protein. Fibronectin protein is secreted and incorporated into the extracellular matrix (ECM), making its detection more challenging in certain cell or tissue samples, such as hypertrophic scars. In contrast, mRNA is typically measured within cells, which may account for the differences in levels detected. Finally, technical variations in measurement methods may also contribute to the discrepancies. mRNA levels were quantified using real-time PCR, while Western blotting was employed to measure protein levels, which may introduce variability between the two (misclassification of measurement errors). We have added this explanation in the discussion of the article [53,54,55,56].

Most of the studies fail due to the inability to blind the researchers during the intervention and also during the assessment. Nevertheless, regarding the ROBINS I tool, most of the studies find a low risk of bias. As mentioned, some outcomes were challenging to compare, and in some instances, only a single article addressed a specific outcome, resulting in a low GRADE score.

#### 3.5.3. Limitations

While human studies provide valuable insights into the mechanisms of SWT in fibrosis, translating these findings into clinical practice is challenging due to the inclusion of in vitro studies. In vitro studies involve different methodologies, such as using a liquid–air interface in cell culture dishes, which may alter the reflection of shock waves [40]. These ideal laboratory conditions contrast with the complexities of clinical settings. Additionally, we decided to separate human and animal studies due to their different results. However, this separation allows us to make human findings more applicable to human practice. Furthermore, less diverse results were reported due to the segregation of human and animal studies.

In the topic, we found several types of studies: fundamental studies and clinical effect studies about pain, redness, and clinical symptoms; there are relevant and important studies that do not contribute to the decision to choose parameters in SWT.

Another limitation is that only one randomized controlled trial (RCT) was found on this research question. Nearly all studies lacked baseline or follow-up assessments. The majority concentrated exclusively on short-term effects, specifically within the first 72 h following SWT.

Additionally, the diversity of populations, such as variations in age, underlying conditions or the severity of fibrosis, can affect how representative the findings are for broader patient groups. Discrepancies in endpoints, such as the criteria for measuring fibrosis resolution or progression, further complicate the synthesis of results. They may limit the generalizability of the findings to different clinical contexts. The need for future research to control for these variables would also strengthen the recommendations for clinical practice. Nevertheless, this systematic review gives insights into certain fibrosis types that respond to therapeutic interventions. A huge strength of this review is that it is the first to comprehensively examine individuals with fibrosis across various tissues, including both internal and musculoskeletal tissues, with overlapping findings. Moreover, the review reports on the different treatment modalities utilized; we made a clinical translation because we examined EFDs underlying mechanisms depending on several types of fibrosis.

Further research is crucial to enhance our understanding of the underlying mechanisms of SWT. We strongly recommend using in vivo models with baseline assessments. It is essential that future studies clearly specify the type and calibration of SWT utilized. Moreover, investigating the long-term effects of shockwave therapy and its potential synergies with pharmacotherapy and physical therapy is necessary. Comparative studies integrating objective and subjective assessments could provide valuable insights.

## 4. Conclusions

Our study has uncovered new insights into the development of non-invasive therapies for tissue fibrosis modulation in humans. SWT stimulates mechanotransduction locally, offering a safe treatment option that effectively modulates the biochemical actions of inflammatory mediators, thereby favorably impacting clinical outcomes. It can be broadly applied in different forms of fibrotic conditions, such as musculoskeletal fibrosis (e.g., post-traumatic tendon injury), dermal fibrosis (e.g., hypertrophic scarring), pulmonary fibrosis, and liver fibrosis [57]. This systematic review has provided a comprehensive analysis of the underlying mechanisms through which shockwave therapy (SWT) influences fibrosis. The evidence indicates that SWT can modulate several key biological processes, including macrophage activation, fibroblast proliferation, collagen organization, and apoptosis, ultimately influencing fibrosis outcomes without significant adverse effects, although isolated reports mention minor reactions such as erythema, petechiae, and edema [18]. The therapeutic potential of SWT is largely dependent on energy levels, frequency, and treatment duration, as these parameters significantly impact molecular pathways such as mechanotransduction. For researchers, this review highlights the need for further exploration into the specific signaling pathways activated by different SWT modalities. While studies have demonstrated promising effects on fibrosis-related proteins and pathways, there remains a lack of consensus on the optimal treatment parameters for different types of fibrotic tissues. Future research should aim to establish standardized protocols, focusing on energy flux density and pulse frequencies, to maximize therapeutic efficacy across various fibrosis types. Additionally, more randomized controlled trials (RCTs) and human-based studies are necessary to bridge the gap between experimental data and clinical practice. For clinicians, SWT emerges as a promising non-invasive treatment for managing fibrosis. The findings suggest that SWT can be integrated into clinical practice as a supplementary therapy for fibrosis, particularly in patients who may not be candidates for more invasive interventions. Clinicians should consider the specific fibrosis type, treatment phase, and individual patient factors when applying SWT, as variations in energy and frequency can lead to differing outcomes. While SWT shows potential for reducing fibrotic tissue and improving patient outcomes, further research will solidify its role in routine care.

In conclusion, SWT represents an innovative approach to treating fibrosis with significant therapeutic potential. By focusing future research on the clinical applicability of SWT protocols and tailoring treatment strategies to patient needs, SWT could become a cornerstone of fibrosis management in both experimental and clinical settings. This review underscores SWT’s potential clinical significance as a key treatment modality for fibrosis across multiple organs. Importantly, SWT has demonstrated safety in early-stage tissue lesions, potentially mitigating pathological remodeling and functional decline in various tissues.

## Figures and Tables

**Figure 1 ijms-25-11729-f001:**
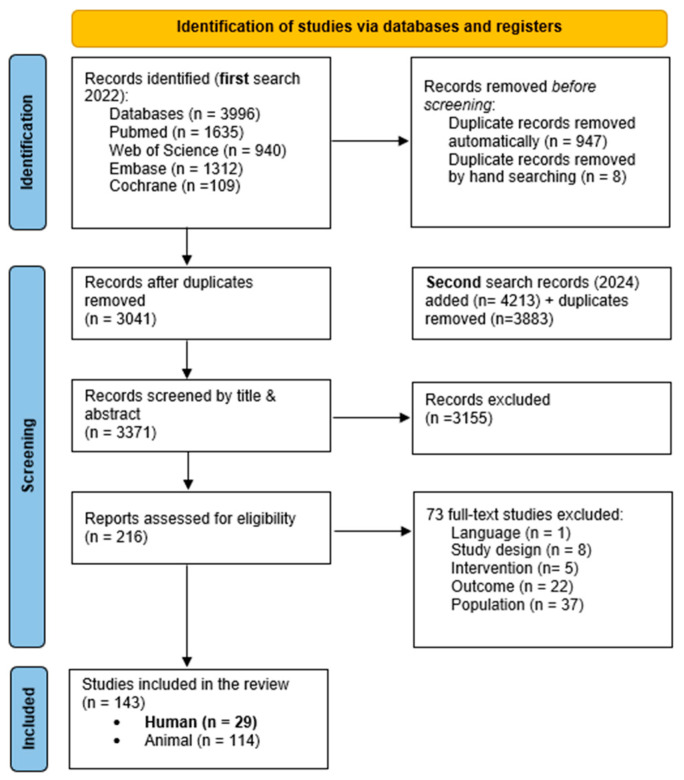
PRISMA flowchart: an overview of the inclusion and exclusion process. Abbreviations: n = numbers.

**Figure 2 ijms-25-11729-f002:**
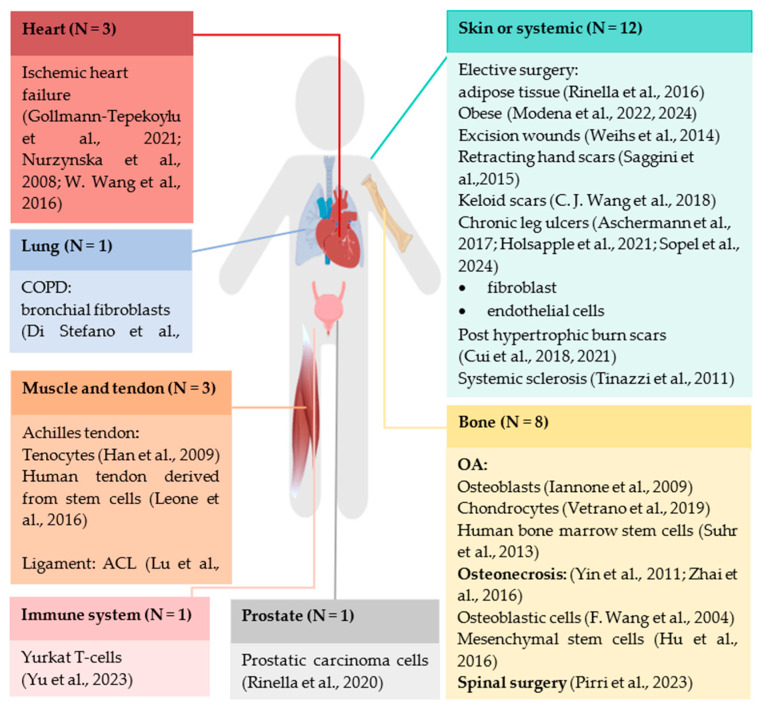
Overview of the included tissues and specific type of fibrosis. In parentheses in the headings the number of articles per tissue (=N) and also the references. Abbreviations: OA = osteoarthritis, ACL = anterior cruciate ligament [3,4,5,6,7,8,9,10,11,12,18,19,20,23,24,25,26,27,28,29,30,31,32,33,37,38,39].

**Figure 3 ijms-25-11729-f003:**
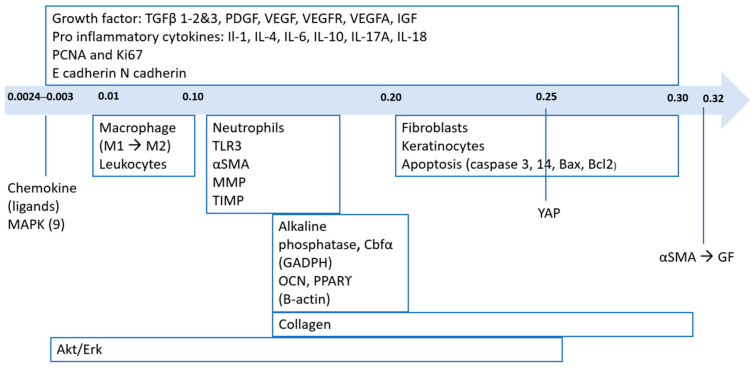
Overview of the energy flux density (mJ/mm^2^) related to the underlying target mechanism. Abbreviations: TGF = transforming growth factor, PDGF = platelet-derived growth factor, VEGF = vascular endothelial growth factor, IGF = insulin-like growth factor, IL = interleukin, PCNA = proliferating cell nuclear antigen, MAPK = mitogen-activated protein kinase, TLR = toll-like receptor, αSMA = alpha-smooth muscle actin, MMP = matrix metalloprotease, Cbfα = core binding factor alpha, GADPH = glyceraldehyde 3-phosphate dehydrogenase, OCN = osteocalcin, PPARϒ = peroxisome proliferator-activated receptor gamma, Erk = extracellular signal-regulated kinase.

**Figure 4 ijms-25-11729-f004:**
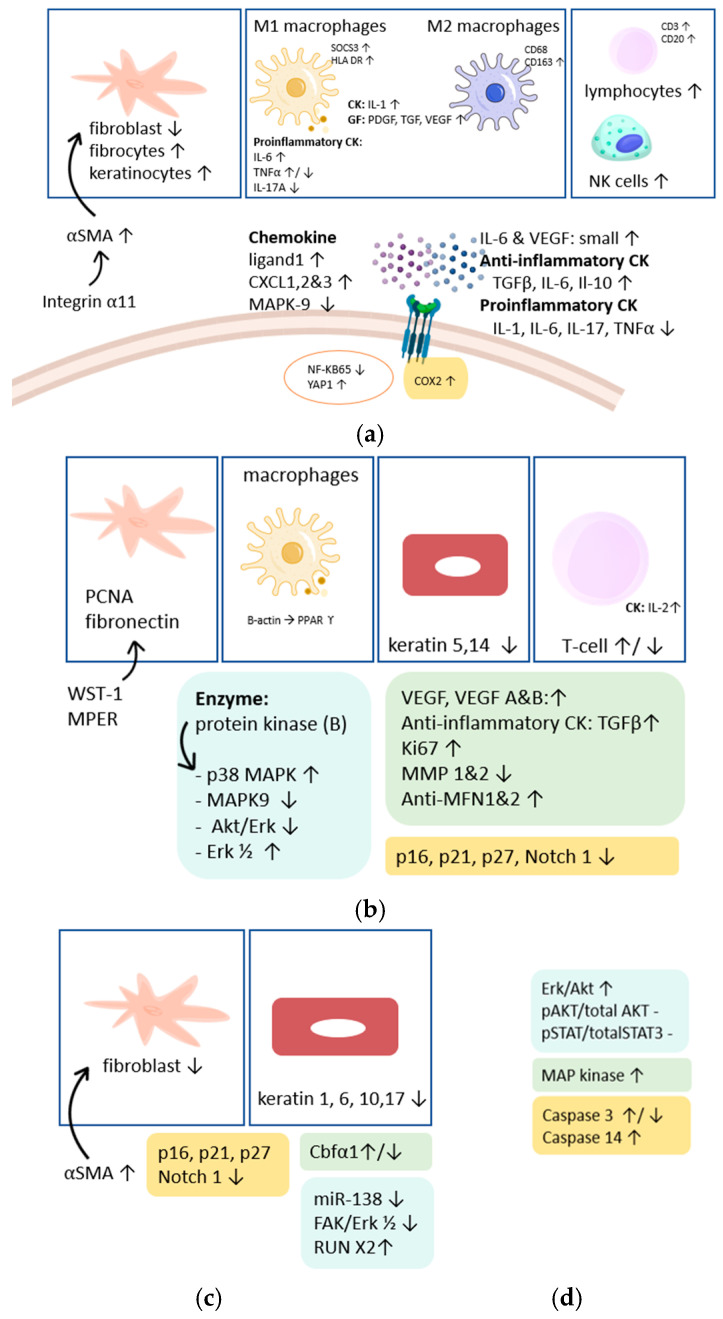
Overview of the effect of SWT on inflammation (**a**), proliferation (**b**), differentiation (**c**) and apoptosis (**d**) phase. Abbreviations: αSMA = alpha-smooth muscle actin, SOCS3 = suppressor of cytokine signaling, HLA DR = human leukocyte antigen-DR, CD = cluster of differentiation, CK = cytokine, GF = growth factor, IL = interleukin, PDGF = platelet-derived growth factor, TGF = transforming growth factor, VEGF = vascular endothelial growth factor, TNFα = tumor necrosis factor, NK = natural killer, MAPK = mitogen-activated protein kinase, YAP = yes associated protein, WST = water-soluble tetrazolium salts, CXCL = chemokine ligand, MMP = matrix metalloprotease, Cbfα = core-binding factor, FAK/Erk = focal adhesion kinase, pSTAT = signal transducers and activators transcription.

**Table 1 ijms-25-11729-t001:** Eligibility criteria related to PICO.

	Inclusion	Exclusion
P	Human subjects Animals In vitro, in vivo, ex vivo analysis Fibrosis	No fibrosis, no tissue damage in human or animal subjects Healthy tissue
I	Any form of shockwave	No shockwave therapy Effects of other therapy next to shockwave therapy
C	/	/
O	Cells and molecules Cellular and molecular pathways Mechanotransduction Mechanisms driving fibrogenesis mRNA, DNA	Not underlying mechanism, all ‘visible/external changes’ due to shock wave therapy
Study design	Other study designs	Systematic reviews, Meta-analysis
Time	/	/
Language	Articles written in English, Dutch, German, French	Articles written in any other language

**Table 2 ijms-25-11729-t002:** Search strategy related to PICO in PubMed.

**Population**
(Fibrosis[MeSH] OR Fibrosis[Tiab] OR pulmonary fibrosis[MeSH] OR Pulmonary fibrosis[Tiab] OR oral submucous fibrosis[MeSH] OR oral submucous fibrosis[Tiab] OR endomyocardial fibrosis[MeSH] OR endomyocardial fibrosis[Tiab] OR Cystic fibrosis[MeSH] OR cystic fibrosis[Tiab] OR connective tissue cells[MeSH Terms]OR connective tissue cells[Tiab] OR skin[MeSH Terms] OR cutaneous tissue[Tiab] OR skin[Tiab] OR subcutaneous tissue[MeSH Terms] OR subcutaneous tissue[Tiab] OR connective tissue[MeSH] OR connective tissue[Tiab] OR fibroblasts[MeSH] OR fibroblasts[Tiab] OR myofibroblasts[MeSH] OR myofibroblasts[Tiab] OR tissue[Tiab] OR tissues[MeSH])
**Intervention**
(“extracorporeal shockwave therapy”[MeSH] OR extracorporeal shock wave*[Tiab] OR extracorporeal shockwave*[Tiab] OR “Lithotripsy”[MeSH] OR lithotripsy[Tiab] OR shockwave*[Tiab] OR high energy shockwave*[Tiab] OR “High-energy shock waves”[MeSH] OR shock wave*[Tiab] OR radial pressure wave*[Tiab] OR acoustic wave*[Tiab] OR ESWT[Tiab] OR HESW[Tiab] OR SWT[Tiab] OR SW[Tiab] OR RPWT[Tiab] OR AWT[Tiab])
**Outcome**
(Mechanotransduction[Tiab] OR Mechanotransduction, cellular[MeSH] OR Mechanoreceptors[MeSH] OR mechanotransduction, cellular[Tiab] OR Mechanoreceptors[Tiab] OR Mechanical signal transduction[Tiab] OR Mechanosensory transduction[Tiab] OR Target cell[Tiab] OR Receptor, cell surface[Tiab] OR Receptors, cell surface[MeSH] OR Signaling pathways[Tiab] OR Signal transduction[MeSH] OR Signal transduction[Tiab] OR Cell signaling[Tiab] OR Signal transduction system[Tiab] OR Receptor mediated signal transduction[Tiab] OR Signal pathways[Tiab] OR Signal transduction pathways[Tiab] OR Cellular[Tiab] OR Cells[Tiab] OR cells[MeSH Terms] OR Cell[Tiab] OR Cell physiology[Tiab] OR Cell physiological phenomena[MeSH] OR Cell physiological process[Tiab] OR Cell Physiological Phenomenon[Tiab] OR Micro RNA[Tiab] OR Micro RNAs[MeSH] OR Micro RNA[Tiab] OR Micro RNAs[Tiab] OR miRNA[Tiab] OR miRNAs[Tiab] OR Exosomes[MeSH] OR Exosomes[Tiab])

*: this truncation symbol at the end of the word will retrieve variations of the ending of the word.

**Table 3 ijms-25-11729-t003:** Evidence table. Overview of studies performing shockwave and the effect.

Skin				Treatment			Evaluation		Conclusion
Underlying Mechanism	Amount	Type of Fibrosis	Reference Study Design	SWT	Other	Timing	Type of Evaluation	Timing	Effect of SWT
Inflammation COX-2 T lymphocyteCD3/B lymphocyte CD20 NK cells Proinflammatory macrophage: CD68 and CD163 Angiogenesis CD34 CD105 VEGF	N = 14 (F)	Obesity: Adipose tissue and skin Pre- operative bariatric surgery	[17] Non-RCT Quasi-experimental design: Post-test design	Left side: frequency: 15 Hz EFD: 180 mJ/4000 shots with 15 mm stainless steel tipEFD: 100 mJ energy/2000 shots with 15-mm plastic tip	Right side: Control	Preop: 2 treatments/ wk	Biopsies: IHC	/	* ↑ expression inflammation (*p* < 0.001): COX-2 CD3 T lymphocyte/CD20 B lymphocyte NK cells * + effects on (*p* < 0.001): CD68/CD163 pro-inflammatory macrophage Angiogenesis markers (*p* < 0.001): CD34: no * difference CD105: * higher VEGF: * higher
Cellular metabolism cAMP Mitochondrial division DRP1 Apoptosis Proliferation Anti-MFN1 Anti-MFN2	N = 8 (F)	Obesity grade I and IIPre- operative pre- preparation of bariatric surgery	[12] Non-RCT Quasi-experimental design: Post-test design	7 SWT sessions EFD: 180 mJ 2000 shots + 15 mm stainless steel tipfrequency: 15 Hz EFD: 100 mJ	/	2 treatments/ wk	Biopsy: IHC	/	cAMP: morphology was maintained -> no * changes (*p* < 0.1823) DRP1: no * positivity for the expression of this protein (*p* < 0.0851) Apoptosis: not reported Anti-MFN1 and anti-MFN2: higher immunopositivity IG → regulation in cell proliferation and MFN2 but not * Anti-MFN1 (*p* < 0.0003) Anti-MFN2 (*p* < 0.4852)
Cell markers CD13, CD44, CD90, CD10, CD14Fibroblast marker αSMACol production: I and V Myofibroblast: TGFβ1 expression Integrin a3,5,v,11	N = 5	Myo- fibroblast of adipose tissue cells → after elective liposuction	[11] Non-RCT Quasi-experimental design: time-series design	EFD: 0.32 mJ/mm^2^ PPP of 90 MPa 1000 pulses frequency: 4 Hz EFD: 0.22 mJ/mm^2^ 1000 pulses// 0.32/1000 pulses// 0.59/250 pulses	ControlTGF β1 and αSMA: different modalities	/	Real-time PCR (gene expression) Immunoblotting Immuno- fluorescence microscopy	0–6–72 h 14 d 21 d	Cell markers: CD14: lower expression CD34-CD45: no expression (in SWT and basal/control conditions) αSMA * ↓ (*p* < 0.001) already after 14 d best ↓ with EFD: 0.33 mJ/mm^2^ Col: I and V * ↓ (*p* < 0.001) Myofibroblast: impaired contractility and migration potential TGFβ1 ↑ → αSMA expression Integrina 11 * ↓ (*p* < 0.001) + effects on wound edges and cell contraction
Collagen fibers Angiogenesis marker: CD31 Vessel abundance: SMC Proliferation marker: Ki67 Macrophages: Activation and number: Markers: CD68 M1–M2: M1: SOCS3 expression, HLA DR/M2: CD163 Macrophage CK and GF: TNF, PDGF, TGF, TNF IL6, VEGF Apoptosis ERK and Akt signaling	N = 9	Chronic ulcers (venous) → not healing > 8 wks	[9] Non-RCT Quasi-experimental design: 1 group pre-post-test design	EFD: 0.1 mJ/mm^2^ frequency: 5 Hz 150–500 impulses	/	/	Pre and post-biopsy—Masson’s trichrome staining	/	↑ but not * Collagen (*p* = 0.618) CD31 (*p* = 0.0391) SMC (*p* = 0.085) Ki67 (*p* = 0.398) CD68 * ↓ M1: HLA DR M2: CD163 expression generally ↑ (60% of the patients) CK and GF: * ↑ TNF (*p* = 0.036) * ↑ IL-1 (*p* = 0.029) PDGF (*p* = 0.029) TGFβ (intensity of 150 pulses)/ (*p* = 0.048) * ↑ TNF & TGFβ (intensity of 300 pulses) No * change of IL6, VEGF (*p* > 0.05) Apoptotic cells ↑ BUT no * changes in pAKT/totalAKT ratios or pSTAT3/totalSTAT3
YAP1Myofibroblast activation: αSMAProliferative status: Ki-67Microvascular density: CD31	N = 10	Pressure ulcers	[10] Non-RCT Prospective interventional pilot study	Radial ESW: Low-energy: 300 pulses, pressure: 2.5 bars EFD: 0.15 mJ/mm^2^, frequency 5 Hz	/	2 sessions, 2 x/wk, 3-day interval in between	Biopsy: IHC assessment and evaluation	baseline and 24 h after first ESW after second ESW	YAP expression: in epidermis and dermis * Differences in YAP expression in keratinocytes (in fibroblasts/myofibroblasts and vascular endothelial cells) Cytoplasm (*p* < 0.001) and nucleus (*p* = 0.0259) YAP1 protein: strong expression after second ESW Myofibroblast activation: αSMA: Baseline and 24 h: lowest values of αSMA After first and second ESW: weak to moderate values of αSMA (*p* < 0.0001) Proliferative index: Ki67: After first and second ESW: * ↑ number of cells for the Ki67: in keratinocytes, dermis cells and the papillary layer of the dermis (*p* < 0.0001) Microvascular density: CD31 After the first and second ESW: the number ↑ (*p* < 0.0001)
In vitro: Morphological changes Cell migration Cell-cycle regulatory genesProliferation Pro-inflammatory CK pro-angiogenic activity of endothelial cells in vivoCellular morphology in fibroblasts, keratinocytes and dermal microvascular endothelial cells	In vivo: N = 75	Chronic leg ulcers: Keratinocytes (HaCaT cells), fibroblasts and endothelial cells in vitro	[3] Non-RCT Quasi-experimental design: time-series design	0, 375, 750, 1500 shocks at energy level Frequency: 4 impulses per s EFD: 0.136 mJ/mm^2^ → 0, 50, 100 and 200 impulses per cm^2^	No treatment	4 times 1 x/wk every 3–4 wks	In vivo: wound healing assessment In vitro: FACs, WB, RT-PCR, asymmetric gene expression analyses	24 h after SW	Cluster formation of fibroblasts dose-dependent Angiogenesis: 375 shocks enough to ↑ capillary formation → cell migration in fibroblasts and cytokines → ↑ wound closure after 24 h due to enhanced migratory capacity → expression of cell cycle regulatory genes and proteinsProliferation: 24 h: no changes in the 72 h: changes in the proliferationAltered expression of cytoskeletal proteins in fibroblasts [reorganization of the cytoskeleton: vimentin]: upregulation after 24 h in WB and dose-dependentActivate immune response factors (in keratinocytes): inflammatory CK: IL-1α upon SWT at 750 impulses and IL-1β at 1500 impulses: * ↑ (*p* < 0.05) Laminin-322 (via collagen VII): SWT: expression of laminin-332 → promoting wound healing
Col I, II, III and X Degrading enzyme: MMP-13 Angiogenesis: CD31, VEGF Anti-inflammatory cytokines: TGFβ1 and IL6 Proliferating and regeneration markers: PCNA and fibronectin Apoptosis markers: Tunel and caspase-3	N = 39 (22 ESWT/17 steroid group)	Keloids scars (>1 cm)	[18] Non-RCT Prospective open-label randomized case study	EFD: 0.11 mJ/mm^2^ 500 impulses. 4 shocks/s 3 ESWT treatments in 6 wks	Steroid group	/	IHC: Masson Trichome stain	6, 12, 24 and 48 weeks after the last treatment	* ↓ in col I and III (*p* < 0.05) * ↑ in MMP-13 (*p* < 0.05) Small but not * changes in angiogenesis, proliferative and regeneration, anti-inflammatory and apoptosis biomarkers
FXII + fibrocytes CD34 of fibroblasts Collagen type I/III CD31 of angiogenesis	N = 70	Retracting hand scars	[19] RCT	10 unfocused ESWT 2 x/wk EFD: 0.13 mJ/mm^2^ Frequency: 6 Hz Duration: 1.5 min (500 pulses for session per 360 impulses/min)	/	/	IHC staining of fibroblasts and angiogenesis	Pre and post-ESWT	* ↑ in dermal fibroblasts, neoangiogenetic response and type I collagen → replacement of type III with type I collagen (*p* < 0.05) * ↑ in fXIIIa-positive fibrocytes, CD34 dermal expression and CD31-positive small vessels (*p* < 0.05)
Fibrosis markers Endothelial involvement: vWF, VEGF, ICAM-1, MCP-1 Circulating endothelial cells (CECs) Endothelial progenitor cells (EPC) Nitric oxide	N = 30 (F = 29/ M = 1)	Systemic sclerosis (excessive deposition of collagen in the skin and visceral organs)	[20] Pilot studyNon-RCT Quasi-experimental design: pre and post-test design	Defocused ESWT Dorsal and volar forearm: 2000 shots Dorsal side of hand and fingers: 1000 EFD: 0.2–0.25 mJ/mm^2^ frequency: 4 Hz	/	/	Blood samples: ELISA	Before, 30 and 60 days after ESWT	Fibrosis marker: No * changes vWF, VEGF, ICAM-1, MCP-1: * ↑ on 30 and 60 days after the end of treatment * ↑ CECs/EPC (*p* < 0.05) NO: ↑ in 2/3 patients (not *)
Proliferation: P38 MAPK P44/42 MAPK Total Akt Total Mek ½	/	Primary human adipose tissue-derived stem cells Full-thickness biopsies from excision wounds	[8] Non-RCT Report Quasi-experimental times series design	In vitro: 10–300 pulses EFD: 0.03–0.19 mJ/mm^2^ 3 Hz In vivo: 100 pulses frequency: 13 Hz Treatment at 0.03, 0.07, 0.13 and 0.19 mJ/mm^2^	Control	/	H&E IHC WB	15 h 24 h 36 h after ESWT	Proliferation and activation: downstream Erk ½, Ki67 activation Mek ½ and P38 MAPK → enhance proliferation 15 h: 0.03 + effects but 0.07 and 0.19 mJ/mm^2^ * enhancement in proliferation [*p* < 0.01] 24 h: 0.03 and 0.07 + effects but 0.19 mJ/mm^2^ * enhancement in proliferation (*p* < 0.05)
TGFβ1 αSMA col type I fibronectin twist I E cadherin N cadherin	N = 4	Human fibroblasts from post-burn hypertrophic scars	[5] Non-RCT Quasi-experimental design: times series design	1000 impulses EFD: 0.1, 0.03 and 0.3 mJ/mm^2^	/	/	Real-time PCR WB For mRNA and protein expression	24 h and 72 h after treatment	TGFβ1, αSMA, vimentin: mRNA and protein: * ↓ after 24 h and 72 h EMT markers: collagen type I mRNA * ↓ after 72 h in all regimens * ↓ after 24 h only in 0.1 and 0.03 mJ/mm^2^ (*p* < 0.05) col type I protein * ↓ after both 24 and 72 h (*p* < 0.05) fibronectin mRNA: 24 h * ↓ in all regimens 72 h: * ↓ only at 0.03 and 0.1 mJ/mm^2^ BUT fibronectin protein was * ↑ 24 h and 72 h after SW with 0.03 and 0.1 mJ/mm^2^ (*p* < 0.05) N and E cadherin (= cell surface markers for EMT) mRNA of N cadherin * ↓ after 24 h and 72 h in all regimens (*p* < 0.05) E cadherin * ↓ after 24 h and 72 h only in 0.1 and 0.3 mJ/mm^2^ (*p* < 0.05)
Proliferation marker: keratin 5 and 14 Activation marker: keratin 6 and 6/17		Keratinocytes form hypertrophic scars (during surgical procedure) Human normal keratinocytes (HNK) Human hypertrophic scar keratinocytes (HTSK)	[4] Non-RCT Quasi-experimental design: times series design	Duolith SD device SD1 Storz medical EFD: 0.1, 0.2, 0.3 mJ/mm^2^ frequency: 4 Hz and 1000 impulses/cm^2^	/	/	mRNA and protein expressionReal-time PCR WB	At 24 h And 72 h	HTSK: Proliferation marker: Keratin 5: no effect Keratin 14: 0.1 mJ/mm^2^: no difference between SW and untreated control at 24 h 0.2–0.3 mJ/mm^2^: * lower in HTSK after 24 h and 72 h [*p* < 0.05] 0.2–0.3 mJ/mm^2^: higher than control cells Activation marker: Keratin 6: At 24 h and 72 h * higher (lower than that of control cells) (*p* < 0.05) Keratin 17: In HTSK 24 h after all SW regimens * higher than control cells (*p* < 0.05) HNK: Protein expression of keratin 6, 14 and 17 in HNKs was unaffected after all SW regimens BUT activation marker keratin 6 expression in HNKs 24 h and 72 h after SW with 1000 impulses/cm^2^ at 0.1, 0.2, 0.3 mJ/mm^2^ * higher and < than untreated cells (*p* < 0.05)
Differentiation marker: keratin 1 and 10 Apoptosis factor: Bax, Bcl2, ASK1, caspase 14 Proliferation and differentiation marker: p21, p27 and Notch1									Differentiation marker: keratin 1: In HTSKs after 24 h and 72 h * lower in Involucrin expression HTSKs (*p* < 0.05) 24 h: 0.1 mJ/mm^2^ not * changed, 24 h: in 0.2 and 0.3 mJ/mm^2^ * higher (*p* < 0.05) 72 h: under all regimens * lower (*p* < 0.05) protein expression of keratin 1 and involucrin in HNKs 24 h and 72 h after SW regimens * higher (*p* < 0.05) Apoptosis factor: Pro-apoptotic factor Bax/Anti-apoptosis factor Bcl2 24 h and 72 h * higher and lower than that of those in untreated control cells (*p* < 0.05) ASK1: in HTSKs 24 h and 72 h after SW * higher (*p* < 0.05) Caspase 14 expression: 24 h after SW was not * different 72 h * ↑ (*p* < 0.05) Proliferation and differentiation marker: p21, p27 and notch 21 0.1 mJ/mm^2^: not * changed 0.2–0.3: * higher at 24 h/at 72 h * lower (*p* < 0.05)
Bone				Treatment			Evaluation		Conclusion
Underlying Mechanism	Amount	Type of Fibrosis	Reference Study Design	SWT	Other	Timing	Type of Evaluation	Timing	Effect of SWT
Osteogenic differentiation: miR-138 differentiation: FAK Erk ½ p38 RUNX2miR138 ERK ½	N = 3 (all male patients)	MSCs [after surgery: resection]; bone marrow, tendon and adipose tissues	[21] Non-RCT Quasi-experimental design: one group pre and post- test design	EFD: 0.16 mJ/mm^2^ 500 impulses	/	/	Real-time PCR WB analysis	/	TG: SW promotes differentiation of MSCs via FAK Marker: RUNX2 ↑ after ESW miR138 * ↓ (*p* < 0.01) ERK ½ downstream in the ERK ½ and p-FAK pathway → enhanced osteogenesis
Collagen type II, βactinProliferation: Ki67Apoptosis: caspase-3Migration Cell tracking in healing, migration and wound healing assay	/	hBMSCs [human bone marrow stromal cells] → femoral head from patients after hip joint replacement	[22] Non-RCT Quasi-experimental design: post-test only control group	SW continuous pulse1000 impulsesFrequency: 4 HzEFD: 0 control 0.2 mJ/mm^2^ 0.3 mJ/mm^2^	control: Sham treatment	/	RT-PCR IHCBoyden chamber assay Cell tracking assay Wound healing assay	6 h and 12 h after SW 0 min and 30 min after SW 4 h and 8 h	Growth rate: both in treated and untreated hBMSCs Proliferation: Ki67 0.2 mJ/mm^2^: * ↑ after 6 h and 12 h (*p* < 0.05) 0.3 mJ/mm^2^: no * effect after 6 h/ * ↓ after 12 h (*p* < 0.05) Apoptosis: caspase-3 0.2 mJ/mm^2^: * ↓ after 6 h/transient effect after 12 h (*p* < 0.05) 0.3 mJ/mm^2^: * ↑ (*p* < 0.05) 0.2 mJ/mm^2^: optimal stimulation, maximal induction of proliferation with minimal * activation of apoptosis (*p* < 0.05) Migration: F-actin stress fibers Disorganized actin fibers of the cytoskeleton: 0 min no changes/30 min: 0.2 mJ/mm^2^ * reduced disorganized F-actin (*p* < 0.05) ↑ wound closure proportion
IL-10 TNF-α CD29/β1 integrin CD105/endoglin expression	N = 13 (9 F/4 M): OA/ N = 7 from HD after joint traumatic fracture	Knee OA Human subchondral bone: osteoblasts	[23] Non-RCT Quasi-experimental design: times series design	EFD: (mJ/mm^2^ and impulses): 0.055 and 500 0.055 and 1000 0.17 and 500 0.17 and 1000	Healthy bone	/	Flowcytometry	/	IL-10 * ↑ in healthy and OA osteoblasts: 0.055 mJ/mm^2^ and 1000 impulses (*p* < 0.05) TNF-α did not vary over time CD29/β1 integrin did not * changed (*p* < 0.05) CD105/endoglin expression: higher by higher EFD (EFD of 0.17 mJ/mm^2^)
VEGF AProtein levels ERK-dependent HIF-1α	/	Fetal preosteoblastic cells	[23] Non-RCT Quasi-experimental design: post-test only control group design	EFD: 0.16 mJ/mm^2^ frequency: 1 Hz 500 impulses	/	Single treatment	RT-PCT immunoblotting	15, 30 and 1,3, 6, 12 and 24 h after SW treatment	VEGF-A * enhanced by SW after 6 h (*p* = 0.015), 12 h (*p* < 0.001) and 24 h (*p* < 0.001) VEGF-A mRNA and protein levels → expression by Ras-induced superoxide (*p* < 0.001) ERK-dependent HIF-1α activation (after 1 h and after 3 h) → Modulation of redox reaction by SW →+ effects on angiogenesis
Cell morphology: SOX-9, COL1A, COL2AProliferation and repair: TNFα, IL-6 SOX-9, COL1A, COL2A, IL-17A Inflammation: IL-10	N = 4	Femoral head (with OA): chondrocytesIn vitro	[24] Non-RCT Quasi-experimental design: post-test only control group design	EFD: 0.14 mJ/mm^2^ Energy level 1000 impulses	Control HA PRP	/	RT-PCRImmuno-fluorescenceImmunoassay kit	12 days after SW treatment	Shift from COL2A to COL1A (collagen type I and II) (*p* < 0.001) COL2A mRNA * higher expression (*p* < 0.05) Enhanced expression of SOX-9 Expression of p16: * ↓ (*p* < 0.05) Ki67: not * changed (*p* < 0.05) Pro-inflammatory cytokines: TNFα, IL-6 * SOX-9, COL1A, COL2A (*p* < 0.001), IL-17A ↓ Anti-inflammatory cytokines: IL-10 * ↑ (*p* < 0.05)
Angiogenesis: VEGFOsteogenesis gene expression: BMp-2 RUNX2Osteocalcin: mRNA expression	N = 6 patients but 4 samples	Osteonecrosis of the femoral head BMSC for hips	[25] Non-RCT Quasi-experimental design: pre-post-test design	250 shockwaves In 2 sessions EFD: 0.18 mJ/mm^2^	Control	/	RT-PCR	72 h after SW treatment	VEGF * ↑ expression in comparison to CG (*p* < 0.05)NOC18 * ↑ the osteogenic markers (BMp-2, RUNX-2) (*p* < 0.05) osteocalcin mRNA * higher expression (*p* < 0.05)
Cell proliferation: CCK-8 OCN (Β-actin) PPARϒ (Β-actin) Alkaline phosphatase Cbfα (GADPH)	N = 20, but only 6 samples	Bone with avascular necrosis of the femoral head (ANFH)	[26] Non-RCT Quasi-experimental design: post-test only control group design	Intervention: 10 min EFD: 0.16 mJ/mm^2^ 500 impulses frequency: 1 Hz MFL 5000 lithotriptor	Control	/	StainingRT-PCR WB	3 d 6 d 9 d 12 d 15 d (after SW treatment)	Gene expression on cell proliferation: CCK-8: * higher in TG then CG (*p* < 0.01) OCN: already detected in 9 d in the ESW group Cbfα1: * higher in ESW over the whole timeline from day 9–15 (*p* < 0.01) PPARϒ: steady for both ESW and control groups at all time points, with significantly weaker levels in the ESW group than in the control group, * results from day 9–15 (*p* < 0.01) Protein expression on cell proliferation: Cbfα1: higher in CG than in TG
Muscle, tendon and ligament				Treatment			Evaluation		Conclusion
Underlying Mechanism	Amount	Type of Fibrosis	Reference Study Design	SWT	Other	Timing	Type of Evaluation	Timing	Effect of SWT
In fibroblasts: YAPProliferation rate	N = 8 (F = 4/ M = 4)	Elective spine surgical procedures from fascia thoracolumbalis	[27] Non-RCT Quasi-experimental design: post-test only control group design	SW EFD: 0.25 mJ/mm^2^ frequency: 2.5 Hz 100 shots	Not treated	/	ImmunoblottingRNA extraction and real-time PCR	2 h 24 h 48 h (after SW treatment)	YAP activation was induced → * difference in treated cells compared to untreated cells (*p* < 0.05) YAP activation → upregulation fibrosis-associated gene expression (COL1A1 and HABP2): COL1A1 * enhanced at 2 h (*p* < 0.0001) 24 h (*p* < 0.05) 48 h (*p* < 0.01) with a peak at 2 h/HABP2 Higher proliferation rates * ↑ at all examined times (*p* < 0.0001)
Proliferation MMP 1-2-9-13 IL 1-6-13Cell proliferationCell death	/	Diseased and fibrotic Achilles tendon → tenocytes	[28] Non-RCT Quasi-experimental design: post-test only control group design	EFD: 0.17 mJ/mm^2^ 250, 500, 1000 and 2000 times	/	/	qPCR	/	In fibrotic tenocytes: proliferation: * ↓ IL-6, MMP1 and MMP13 (*p* < 0.05) MMP2 and MMP9 did not * change * ↑ IL-1 (*p* < 0.05) Cell proliferation: 500 shock group no * diff/1000 2000 shock group: * ↓ Cell death: 500 and 1000 shock group * ↑ (*p* < 0.05)
Differentiation marker: αSMAFibroblast marker: CD 90.2Mesenchymal stem markers: CD44, CD105 and CD146	N = 5 healthy (ST) N = 5 ruptured (AT)	Human tendon-derived stem/progenitor cells (hTSPC) From Achilles tendon and semitendinosus tendon	[29] Non-RCT Quasi-experimental design: post-test only control group design	/	/	/	Immuno-fluorescenceRT-PCR	/	* ↑αSMA ST (*p* < 0.05), ↑ in AT but not * ESWT effect: Ki67—analysis of a nuclear marker: after 12 daysUpregulation of COL2A and SOX9 BGLAP and RUNX2 not * different
Cell viability, proliferation, migration AND expression of Ki67, COL-I α1, TGFβ, and VEGF	N = 8 (M = 5/ F = 3)	Anterior cruciate ligament	[26] Non-RCT Quasi-experimental design: post-test only control group design	1000 SW impulses EFD: 0.15 mJ/mm^2^	Control group: no SW	/	Real-time PCR Scratch migration assay Transwell migration assay Immuno-fluorescence staining	0–32 hAfter 12 days: Ki67	* ↑ cell viability, proliferation and migration (Ki67) (*p* < 0.01) ↑ COL-I α1, TGFβ, and VEGF expression BMSC proliferation and migration rate * ↑ after coculture [with ACL remnant cells with and without ESW stimulation] (*p* < 0.01)
Lung				Treatment			Evaluation		Conclusion
Underlying Mechanism	Amount	TYPE OF FIBROSIS	Reference Study Design	SWT	Other	Timing	Type of Evaluation	Timing	Effect of SWT
Bronchial fibroblasts CD117 PCNA CD90 TGFb1 Procollagen 1 NF-kB-p65	N = 6 (N = COPD/ N = 3 control)	COPD Bronchial fibroblastsCOPD patients	[30] Non-RCT Quasi-experimental design: post-test only control group design	Piezoelectric EFD: 0.3 mJ/mm^2^ frequency: 4 shocks/s 500 pulses	Control: non-treated fibroblasts	/	qRT-PCR ELISA IHC	0, 4, 24, 48 and 72 h (after SW treatment)	24 and 48 h: ↑ proliferation mRNA protein levels remodelling markers PCNA mRNA: not ↑ in the ESW group
Cell proliferation: WST-1 M-PER		Control smokers with normal lung function							PCNA protein: Tendency to ↑TGFb1 mRNA not * changed/TGFb1 protein * ↑ (*p* < 0.01: 24 h and *p* < 0.001: 48 h)Procollagen I: no * difference between TG and CGNF kB-65 * ↓ (*p* < 0.01: 24 h and *p* < 0.001: 48 h) WST-1 and M-PER: cell proliferation and * ↑ of specific markers of cell proliferation (*p* < 0.01: 24 h and *p* < 0.001: 48 h)
Heart				Treatment			Evaluation		Conclusion
Underlying Mechanism	Amount	Type of Fibrosis	Reference Study Design	SWT	Other	Timing	Type of Evaluation	Timing	Effect of SWT
VEFGA VEGFB Chemokine ligan1 CXCL1,2,3 TNF α MAPK-9	N = 23 (M = 17/ F = 6)	Ischemic heart failure Intervention study	[31] Non-RCT Quasi-experimental design: pre and post-test design	Cardiac shock wave therapy with an electromagnetic shockwave device 9 sessions (200 shocks/session) EFD from 0.0024 to 0.09 mJ/mm^2^	/	3 times per week for 3 weeks	qPCR	3 days before CSWT (baseline) and 1 week after CSWT	CSWT: 12 pathways were upregulated 5 pathways were downregulated + correlation with cytokine and cytokine receptor and chemokine pathwayVEGFA, VEGFB, chemokine ligand 1, CXCL1,2 and 3, TNFα * ↑ (*p* < 0.0001) MAPK-9: * ↓ (*p* < 0.0001)
miR-19a-3pCD9, CD63 and CD81	/	Isolation of HUVECs and myocardial infarctionIn vitro	[6] Non-RCT Quasi-experimental design: post-test only control group design	/	/	day 1 and 3 SWT	Flow cytometry	2 min, 30 min, 1 h, 4 h and 24 h after SWT	Proliferation and angiogenesis Akt inhibition: by CD9, CD81 and CD63/Erk → enhanced endothelial and proliferation + αSMA CD9, CD63 and CD81 * higher (*p* < 0.01) miR-19a-3p: induces inhibition of TSP-1 mRNA translation and facilitates the angiogenic effect
Neo-angiogenesis: VEGFR-2αSMA α/βMHC, FVIII and GADPH Apoptosis	N = 16 (M = 9/F = 7) healthyN = 8 (M = 5/F = 3) cardiomyopathy patients	In vitro: cardiomyocytes, smooth muscle and endothelial lineage > normal heart > cardiomyopathy	[32] Non-RCT Quasi-experimental design: post-test only control group design	800 impulses PPP: 18 MPa EFD: 0.10 mJ/mm^2^	No SW		Immuno-fluorescenceTUNEL assayWBRT-PCR	7 d cultured	In neo-angiogenesis: * ↑ in VEGFR-2 expression (*p* < 0.05) * ↑ αSMA (*p* < 0.05) Effect on transcriptional activity: mRNA with primers: α/βMHC, FVIII and GADPH housekeeping: * upregulated expression (*p* < 0.05) Proliferation and differentiation in cardiac primitive cells Apoptosis: no statistically significant difference in both groups
Prostate				Treatment			Evaluation		Conclusion
Underlying Mechanism	Amount	Type of Fibrosis	Reference Study Design	SWT	Other	Timing	Type of Evaluation	Timing	Effect of SWT
Cell viability Gene expression: αSMA COL1	N = 10 tumor tissue samplesIn vitro	Human prostatic cancer cells: Carcinoma associated	[33] Non-RCT: Quasi-experimental design: post-test only control group design	SWT1: CAF E8-1000 EFD: 0.32 mJ/mm^2^, 1000 pulses frequency = 4 shocks/s	/	/	/	72 h 48 h (after SW treatment)	ESW effect on: Cell viability: no sign effect/ Small reduction αSMA * ↓: SWT2 (26%↓) > SWT1 (4.3%↓) (*p* < 0.001) COL 1 * ↓: SWT1 (51%↓) > SWT2 (45%↓) (*p* < 0.001)
Growth of epithelial cells		fibroblasts (CAFs) After prostatectomy		SWT2: CAF E12-250: EFD: 0.59 mJ/mm^2^ 250 pulses frequency: 4 shocks/s PPP: 64 MPa				Protein levelsWB Immuno- fluorescenceMatrigel invasion assay + scratch wound assay	* ↓ αSMA and COL1 (*p* < 0.01) * ↓ PC3 and DU145: and * ↓ in migration (*p* < 0.001)Both energy levels: effective in reducing COL 1 and αSMA BUT E12-250: higher ↓ αSMA
Immune system				Treatment			Evaluation		Conclusion
Underlying Mechanism	Amount	Type of Fibrosis	Reference Study Design	SWT	Other	Timing	Type of Evaluation	Timing	Effect of SWT
Cell stimulation Cell viability ATP release Transfection: siRNA: to block FAK expression T-cell proliferation	N = 3	Jurkat T-cells	[7] Non-RCT: Quasi-experimental design: post-test only control group design	frequency: 50 Hz PPP: 23+-1.4 MPa EFD: 0.18 mJ/mm^2^ Exposed to 0, 50, 100, 150, 200, 250, 300, 350, 400, 450, 500 or 600 impulses Time: 10–20 min	Control	/	WB analysis ELISA		phosphorylation of FAK on Tyr397 or Tyr 576/577 P38 MAPK on Thy180/Tyr182: * ↑ with higher impulses till 250 BUT * ↓ with higher impulses + baseline reached with 350 impulses FAK activation -> p38 MAPK 100 pulses: *p* < 0.01 200 pulses: *p* < 0.01 300 pulses: *p* < 0.001 T-cell proliferation by upregulation of IL-2 expression in a p38 MAPK

Significance is indicated with *, the accompanying *p*-value is indicated per study; / = not mentioned; ↑ = increase; ↓ = decrease; Abbreviations: COX-2 = cyclooxygenase, CD = cluster of differentiation, NK = natural killer, VEGF = vascular endothelial growth factor, RCT = randomized controlled trials, EFD = energy flux density, IHC = immunohistochemistry staining, cAMP = cyclic adenosine monophosphate, αSMA = alpha-smooth muscle actin, Col = collagen, TGF = transforming growth factor, Hz = hertz, PPP = peak positive pressure, SMC = smooth muscle cells, HLA DR = human leukocyte antigen, CK = cytokine, GF = growth factor, TNF = tumor necrosis factor, PDGF = platelet derived growth factor, IL6 = interleukin-6, ERK, Akt, YAP= Yes-associated protein 1, WB = western blot, RT-PCR = reversed transcription PCR, SW = shockwave, MMP-13 = matrix metalleproteases, TUNEL = dUTP nick-end labeling, vWF = Von Willebrand factor, ICAM-1 = intercellular adhesion molecule, MCP = monocyte chemoattractant protein, CEC = circulating endothelial cell, EPC = endothelial progenitor cells, MAPK/MEK = mitogen-activated protein kinase, H&E = Hematoxylin and Eosin, EMT = epithelial mesenchymal transition, Cbf = core binding factor, OCN = osteocalcin, PPAR = proliferation-activated receptor, COPD = chronic obstructive pulmonary disease, NF-kB-p65 = nuclear kappa factor B, WST = water soluble salts, CXCL = chemokine ligand, qPCR = quantitative PCR, CSWT = cardiac shockwave therapy, GADPH = glyceraldehyde-3-phosphate dehydrogenase.

**Table 4 ijms-25-11729-t004:** GRADE score.

Outcome	N	Within Study Bias	Reporting Bias	Indirectness	Imprecision	Heterogeneity	Incoherence	Confidence Rating
Cell contraction	5							Very low
Fibroblasts	7							Very low
Myofibroblasts	3							Very low
αSMA	4							Very low
Fibrocytes	1							Very low
ECM component	2							Very low
Stromal cells	1							Very low
Collagen I	7							Low
Collagen II	3							Very low
Collagen III	2							Low
Collagen V	1							Low
Collagen marker	3							Low
Endothelial cells	1							Low
Nitric oxide	1							Low
cAMP	1							Very low
Keratinocytes	2							Low
Macrophages	2							Low
Cytokines	4							Low
Chemokines	1							Low
Lymphocytes	1							Low
NK cells	1							Low
COX2	1							Low
Angiogenesis marker	3							Low
Growth factor	7							Low
miR-19a-3p	1							Very low
Proliferation	5							Very low
Protein kinase	3							Low
MAPK, Mek	2							Low
Cell surface receptors	2							Low
Alkaline phosphatase	1							Low
Anti-MFN	1							Very low
FAK/Erk	1							Low
Cbfα1	1							Low
Erk, Akt, total Akt	1							Low
pSTAT3	1							Low
Caspase 3,14	2							Very low

**Table 5 ijms-25-11729-t005:** Overview of the effects of shockwave therapy on underlying mechanisms and during different stages.

Not Specified in Stage of Fibrosis				
Cell Type	Cell Category/Marker	Mediator/Gene Expression	Cluster of Differentiation	EFFECT of Shockwave
Myofibroblasts	Cell contraction and migration	/	/	impaired contractility and * increased migration potential [11]
	GF ECM Integrin	TGF β1α11		↑ à αSMA [11] * ↓ [11] (*p <* 0.001)
Fibroblasts	/	Fibronectin: mRNAFibronectin: protein	/	Amelioration: * ↓ [5] (*p <* 0.05) Amelioration: * ↑ [5] (*p <* 0.05)
Osteoblasts	/	β integrin	CD29	not * changed [23] (*p <* 0.001)
Bone marrow stromal cells	/	F-actin		0.2mJ/mm^2^ * ↓ disorganized F-actin [22] (*p <* 0.05)
CollagenEMT marker	I	/	/	↓ [11]/ * ↓ [18] (*p <* 0.05)/ ↓ [33]
	I	mRNA/protein	/	* ↓ [5] (*p <* 0.05)/↑ [38]
	COL-I α1	/		* ↑ [27] (*p <* 0.05)
	II	/	SOX9	no differences in TG and CG [22] upregulation [29]
	I/II	/	/	shift from COL 1A à COL 2A (regulated by enhanced SOX9) [24]
	III	/	/	* ↓ [18] (*p <* 0.05)
	I/III	/	/	replacement of collagen type III with type I [19]
	V	/	/	↓ [11]
Cell expression	TGFβ1	/	BMp-2 RUNX2	* ↑ [25] (*p <* 0.05) * ↑ [25] (*p <* 0.05)
Circulating endothelial cells	CECs	/	/	* ↑ [20] (*p <* 0.05)
Endothelial progenitor cells	EPC	/	/	* ↑ [20] (*p <* 0.05)
Nitric oxide	NO	/	/	↑ in 2/3 patients (not * ) [20]
Growth factor	GF	VEGF	/	* ↑ [20] (*p <* 0.05)
Fibrosis markers Endothelial involvement	Immunoglobulin	/	vWF, ICAM-1, MCP-1	* ↑ [20] (*p <* 0.05)
Cellular metabolism	cAMP		/	no * change [12]
Inflammation				
Cell type	Cell category/marker	Mediator/Gene expression	Cluster of differentiation	EFFECT of shockwave
Fibroblasts	/ PCNAGF Procollagen I NF-kB-p65	/ mRNA protein TGFβ1 mRNA protein / /	CD117, CD90	↑ [30] not ↑ tendency to ↑ not * changed [30] ↑ [30] no * difference [30] ↓ [30]
Fibroblasts/myofibroblasts	YAP1 protein	/	/	* expression [10,27]
	αSMA	/	/	higher values of αSMA after the first and second treatment in comparison to baseline [10]
	fibrocytes	/	CD34	* ↑ [19] (*p <* 0.05)
Keratinocytes	CK/GF	IL-6/VEGF	/	change but not * [18]
Macrophages	/ M1-M2 macrophages	/	CD68 M1: SOCS3 expression, HLA DRM2: CD163	* ↓ [9] (*p <* 0.05) ↑ expression [9] ↑ expression [9]
Macrophages	CK GF CK/GF	IL-1 TNF, PDGF, TGF IL-6/VEGF	/ / /	* ↑ [9] (*p <* 0.05) * ↑ [9] (*p <* 0.05) no * change [9]
/	/	mRNA	CD44	* ↑ [24] (*p <* 0.001)
/	Pro-inflammatory CK	TNFα IL-6 IL-17A	/ / /	* ↓[24] (*p <* 0.001) [23] (*p <* 0.05) ↑ [31] * ↓ [24] (*p <* 0.001) [28]
Lymphocytes	T-lymphocytes B-lymphocytes	/ /	CD3 CD23	higher expression [17]
NK cells	/	/	/	↑ expression [17]
/	Chemokine pathway	/	ligand 1 CXCL1,2 and 3	↑ [31]
/	Chemokine	/	MAPK-9	↓ [31]
Enzymes	/	COX2	/	↑ expression [17]
Integrin α11	/	/	/	↓ overexpression [11]
Angiogenesis				
Cell type	Cell category/marker	Mediator/Gene expression	Cluster of differentiation	EFFECT of shockwave
Myofibroblasts	/	/	CD34	no * difference [17]/no expression [11]
	/	/	CD14	lower [11]
Angiogenesis markers	/	/	CD105 CD105/endoglin	not * changed expression [17]
/	GF/ /	VEGFVEGFA VEGFR-2	// /	* higher [3,17,25] *p <* 0.001 enhanced by Ras-Erk-HIF-α [41] * ↑ [32]
/	αSMA	/	/	↑ [32] ↓ [33]
Collagen fibers	/	/	CD31	* ↑[9,19] (*p <* 0.05) more ↑ after a first and a second treatment [10]
/	/ /	miR-19a-3p /	CD9 CD63 CD81	inhibition of TSP-1 mRNA translation [6]
Fibroblasts	/	Laminin/integrin	/	ameliorated due to SWT [3,43]
Proliferation				
Cell type	Cell category/marker	Mediator/Gene expression	Cluster of differentiation	EFFECT of shockwave
/	PCNA & fibronectin	/	/	small but not * change [18]
/	GF	TGFβ	/	↑ [38]
/	/	VEGF VEGF A and B	/ /	↑ [38] ↑ [31]
/	Ki67	/	/	depending on the EFD: [22] (*p <* 0.05) 0.2: * ↑ after 6 h and 12 h 0.3: no * effect after 6 h/ * ↓ after 12 h not * changed [24] enhanced [8] * ↑ [29] (*p <* 0.05) more ↑ after a first and second treatment [10]
Enzyme	Protein kinase	P38 MAPK	/	enhanced [8]
		P44/42 MAPK	/	/
		MAPK9	/	↓ [31]
	Protein kinase B	Total Akt: Akt/erk Erk1/2	//	downstream regulation [8] enhanced [43] (through VEGF and VEGF-R)
	/	Total Mek 1/2	/	downstream regulation [8]
Epithelial cells	Keratin5, 14	/	/	depending on the EFD: 0.1: no effect/but 0.2–0.3: * lower [4]
Cell surface receptors	/	/	p21, p27 and Notch1 p16	depending on the EFD: 0.1: not * changed [4] (*p <* 0.05) 0.2–0.3: 24 h * higher/72 h * lower [4] (*p <* 0.05) * ↓ [24] (*p <* 0.05)
Isoenzymes	/	/	Alkaline phosphatase	higher in ESW than the control group [26]
	GADPH	/	Cbfα	higher in ESW over the whole timeline [26]
	Β-actin	/	OCN	detected already in 9 d (not in the control group) [26]
Macrophages	Β-actin	/	PPARϒ	steady for both ESW and control groups at all time points, with significantly weaker levels in the ESW group than in the control group [26]
Fibroblasts	/		WST-1	↑ [30]
	/		M-PER	↑ [30]
T-cell	/ /	/ /	Through ATP, P2X7 receptors FAK activation and MAPK	↑ with higher impulses till 250 BUT ↓ with higher impulses + baseline reached with 350 impulses [7]
	CK	IL-2 expression	/	enhance IL-2 expression [7]
ECM remodelling and cell adhesion	Enzyme	/	MMP1 and 2	* ↓ [28]
	/	mRNA	α/βMHC, FVIII and GADPH	upregulated expression [32]
	Anti-MFN1 Anti-MFN2	/ /	/ /	higher scores [12]
Activation/differentation marker				
Cell type	Cell category/marker	Mediator/Gene expression	Cluster of differentiation	EFFECT of shockwave
Keratinocytes	Keratin 6, 17	/	/	* higher [4]
Keratinocytes	Keratin 1, 10	/	/	depending on the EFD [4] 24 h: 0.1 mJ/mm^2^ not * changed/in 0.2 and 0.3 mJ/mm^2^ * higher 72 h: under all regimens * lower
	/	/	p21, p27 and Notch1	depending on the EFD: 0.1: not * changed [4] 0.2–0.3: 24 h * higher/72 h * lower [4] (*p <* 0.05)
	/	/	miR-138 differentiation: FAK/Erk ½ pathway RUNX2	* ↓ [21] (*p <* 0.01) downstream ↑
	/	/	Cbfα1: Gene expression Protein expression	* ↑ [26] (*p <* 0.01) * ↓ [26] (*p <* 0.01)
	αSMA	/	/	↑ [29]
Apoptosis				
Cell type	Cell category/marker	Mediator/Gene expression	Cluster of differentiation	EFFECT of shockwave
Apoptotic cells	Erk Akt signaling	/	/	↑ [9]
	/	/	pAKT/totalAKT	no changes [9]
	/	/	pSTAT3/totalSTAT3	no changes [9]
	Caspase 3	/	/	small but not * change [18]
	Caspase 3	/	/	depending on the EFD: 0.2 mJ/mm^2^: * ↓ after 6 h/transient effect after 12 h 0.3 mJ/mm^2^: * ↑ (*p* < 0.05) [22]
	Caspase pathway	/	Bax, Bcl2	* higher (but lower than the untreated cells) [4]
	Caspase 14	/		24 h not * different/72 h * higher [4]
Protein kinase	MAP kinase	/	Ask1	* higher after 24 h and 72 h [4]

Significance is indicated with *, the accompanying *p*-value is indicated per study / = not mentioned; ↑ = increase; ↓ = decrease; Abbreviations: GF = growth factor, ECM = extracellular matrix, TGF = transforming growth factor, mRNA = fibronectin messenger ribonucleic acid, COL = collagen, SOX = SRY-related HMG box, TG = treatment group, CG = control group, CEC = circulating endothelial cell, EPC = endothelial progenitor cells, NO = nitric oxide, vWF = Von Willebrand factor, ICAM = Intercellular adhesion molecule, MCP = monocyte chemoattractant protein, cAMP = cyclic adenosine monophosphate, PCNA = proliferating cell nuclear antigen, NF-kB-p65 = nuclear kappa factor B-p65, CD = cluster of differentiation, YAP = Yes-associated protein, αSMA = alpha-smooth muscle actin, CK = cytokine, IL = interleukin, VEGF = vascular endothelial growth factor, TNF = tumor necrosis factor, PDGF = platelet derived growth factor, CXCL = chemokine ligand, MAPK/MAP = mitogen-activated protein kinase, COX = cyclooxygenase, SWT = shockwave therapy, Cbf = core binding factor.

## Data Availability

Data can be found within the article.

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
