# Peer review of "Systematic Review on Working Mechanisms of Signaling Pathways in Fibrosis During Shockwave Therapy"

_ijms, 2024, doi:10.3390/ijms252111729_

Round 1

Reviewer 1 Report

Comments and Suggestions for Authors

In this manuscript authors systematically reviewed the literature to elucidate the underlying effect of SWT on fibrosis.  The studies discussed by the authors revealed that, depending on the energy levels and frequency of SWT, other proteins and pathways can be activated demonstrating that SWT has beneficial effects on fibrosis by influence on the proteins and pathways. 

The manuscript is interesting, quite well written and the topic is new. Tables should be simplified but figures are clear. However, several points must be improved. In particular:

Introduction: Authors should list the main factors involved in fibrosis such as cytokines (e.g. TGFB).

Line 282, 270 and others: References must be insert according to the journal style

Lines 403-406: How do the authors explain these contrasting results between fibronectin mRNA and protein levels? May be due to the plasticity in degradation (by proteases) and syntesis of this protein? since it plays a key role in normal tissue remodelling (PMID: 28076935)

Line 428: correct  NF-kB-65 with  NF-kB-p65

Shockwave therapy increases angiogenesis: The role of angiogenesis in fibrosis deserves to be added since inflammation (a key player in fibrotic process) plays a key role in angiogenesis (see PMID: 37443812 ). 

Tables should be formatted according to the journal style and, when possible, should be simplified. 

An accurate revision of punctuation, syntax and abbreviatons used is recommended (see line 434, NF kB-65)

Figure 4 must be moved in a single page

Abbreviations must be written in full lenght when mentioned for the first time

Author Contributions: Lines 688-689 and 694-695 are a leftover of the journal template, remove 

Reviewer 2 Report

Comments and Suggestions for Authors

Major Comments:

1. The manuscript provides a comprehensive overview of shockwave therapy (SWT) and its potential in fibrosis treatment. However, the discussion on the mechanistic pathways, especially mechanotransduction, remains somewhat general. 

2. While the review highlights the potential benefits of SWT on fibrosis, it lacks a clear connection to clinical applications. 

3. The manuscript acknowledges the predominance of in vitro and animal studies but does not adequately critique the limitations of relying on these models.

4. The studies included in the review use a wide range of energy flux densities (EFDs) and pulse frequencies for SWT. The manuscript should elaborate on how these variations affect outcomes and suggest standardized dosages for specific types of fibrosis. 

5. Although SWT is presented as a promising therapy for fibrosis, there is little discussion of how it compares to other treatment options such as pharmaceutical interventions, surgery, or other physical therapies like ultrasound. 

6. The review includes a variety of studies with differing designs, populations, and endpoints. A more robust discussion of this heterogeneity, particularly how it impacts the generalizability of the findings, would add depth to the review. 

Minor Comments:

1. The manuscript uses several abbreviations like SWT, EFD, and TGF-β without defining them clearly at their first mention. Adding a list of abbreviations or ensuring that they are fully spelled out upon their initial appearance would improve readability.

2. Minor grammatical and typographical errors are present throughout the manuscript. For example, terms like “mechanotransduction” are sometimes inconsistently used. A thorough proofreading is recommended to enhance the overall quality and professionalism of the manuscript.

3. The references section contains minor inconsistencies in formatting. Ensuring uniformity in citation style according to the journal’s guidelines will give the manuscript a more polished and professional appearance.

4. The conclusion of the manuscript could be more robust. Summarizing the key findings of the review and offering clear take-home messages for both researchers and clinicians would provide a stronger and more cohesive ending to the paper.

Round 2

Reviewer 1 Report

Comments and Suggestions for Authors

the manuscript can be accepted in the current form

Reviewer 2 Report

Comments and Suggestions for Authors

No more comments

Comments on the Quality of English Language

The English could be improved to more clearly express the research